Taxonomic revision of Eoalligator (Crocodylia, Brevirostres) and the paleogeographic origins of the Chinese alligatoroids

Wang Yan-yin paleolacerta@gmail.com 1 2
Sullivan Corwin 1
Liu Jun liujun@ivpp.ac.cn 1
1 Key Laboratory of Vertebrate Evolution and Human Origins of Chinese Academy of Sciences, Institute of Vertebrate Paleontology and Paleoanthropology , Beijing , China
2 University of Chinese Academy of Sciences , Beijing , China
Young Mark
Electronic publication date: 2016 Aug 24
Publication date: 2016
Volume: 4
Electronic Location ID: e2356
Received 2016 Mar 20; Accepted 2016 Jul 22
Copyright: ©2016 Wang et al.
Copyright year: 2016
Copyright holder: Wang et al.
License: This is an open access article distributed under the terms of the Creative Commons Attribution License, which permits unrestricted use, distribution, reproduction and adaptation in any medium and for any purpose provided that it is properly attributed. For attribution, the original author(s), title, publication source (PeerJ) and either DOI or URL of the article must be cited.
License URL: https://creativecommons.org/licenses/by/4.0/

Keywords: Eoalligator, Protoalligator, Alligatoroidea, Crocodyloidea, Phylogeny, Taxonomy, Paleobiogeography

Funding: National Natural Science Foundation of China 41172017 41472017 IRG-China/South Africa Research Cooperation Programme 95449 This research was funded by National Natural Science Foundation of China (41172017 and 41472017), and IRG-China/South Africa Research Cooperation Programme (No. 95449). The funders had no role in study design, data collection and analysis, decision to publish, or preparation of the manuscript.

==============================
Background. The primarily Neotropical distribution of living alligatoroids raises questions as to when and how the ancestors of Alligator sinensis migrated to China. As phylogeny provides a necessary framework for historical biogeographic issues, determining the phylogenetic positions of the Chinese alligatoroids is a crucial step towards understanding global alligatoroid paleobiogeography. Besides the unnamed alligatoroids from the Eocene of Guangdong Province, three Chinese fossil taxa have been referred to Alligatoroidea: Alligator luicus, Eoalligator chunyii and Eoalligator huiningensis. However, none of these fossil taxa has been included in a phylogenetic analysis. The genus Eoalligator was established to accommodate E. chunyii from Guangdong Province. E. huiningensis from Anhui Province was later erected as a second species, despite no distinctive similarities with E. chunyii. By contrast, the putative crocodyline Asiatosuchus nanlingensis was established based on material from Guangdong Province, close to the E. chunyii specimens geographically and stratigraphically. Furthermore, specimens of A. nanlingensis and E. chunyii share four distinctive characters, but display no evident differences. As a result, the taxonomic relationships of these three species require restudy.

Methods. In this paper, all specimens of E. chunyii and E. huiningensis are reassessed in detail, and compared to specimens of A. nanlingensis. Detailed re-descriptions and revised diagnoses are provided, and a cladistic analysis is carried out to assess the phylogenetic positions of E. chunyii, E. huiningensis and A. nanlingensis.

Results. The analysis recovers E. chunyii and A. nanlingensis as sister taxa among basal Crocodylidae, while P. huiningensis is posited as an alligatoroid. Two key characters support the monophyly E. chunyii + A. nanlingensis: sulcus within surangular, and anteroposteriorly oriented surangular-articular suture. The former character is unique to E. chunyii and A. nanlingensis among crocodyloids, although a smaller and more posteriorly positioned surangular fossa is known in Diplocynodon. Detailed comparisons show the two species to be synonymous, with E. chunyii as the junior synonym of A. nanlingensis based on page priority. Because E. chunyii was erected as the type species of Eoalligator, the genus is now invalid. We establish the new genus Protoalligator to accommodate “Eoalligator” huiningensis, an alligatoroid whose exact phylogenetic position is uncertain. In particular, P. huiningensis retains primitive characters such as a lacrimal that extends further anteriorly than the prefrontal, and a notch at the premaxilla-maxilla suture. However, P. huiningensis also appears to share one important derived character, a complete nasal bar, with alligators. Our taxonomic revisions imply that four alligatoroids are currently known from China, and these species must have dispersed from North America to Asia in more than one event.

Introduction

Crocodylia is a group of reptiles that contains at least twenty-four extant species and is divided into three major clades: Gavialoidea, Crocodyloidea and Alligatoroidea. Alligatoroidea contains the two clades Diplocynodontinae and Globidonta, the second of which consists of Alligatorinae and Caimaninae. Caimaninae includes six extant species, whereas Alligatoridae includes only two extant species: Alligator sinensis (Fauvel, 1879) from China and Alligator mississippiensis (Daudin & Sonnini, 1801) from the southeastern United States. To clarify the historical biogeography of Alligatoroidea, it is necessary to first evaluate the phylogenetic relationships among fossil and extant members of this clade.

Alligatoroidea is low in extant diversity compared to many other reptile groups, but the alligatoroid fossil record contains much greater taxonomic richness. Around the world, fossil alligatoroids are known from Asia, North and South America, and Europe. The oldest known members of the clade, Leidyosuchus canadensis and Deinosuchus riograndensis, are both from the Upper Cretaceous of North America (Lambe, 1907; Rivera-Sylva et al., 2011). The crown group of Alligatoroidea, namely Alligatoridae, is defined as the last common ancestor of Alligator mississippiensis and Caiman crocodilus (Linnaeus, 1758) plus all of its descendants. All known alligatorids come from Asia and the Americas, apart from two extinct basal European species: Hassiacosuchus haupti (Weitzel, 1935) and Arambourgia gaudryi (Stefano, 1905). More broadly, most fossil alligatoroids are from North America, which may represent an evolutionary center from which alligatoroids have repeatedly dispersed to other continents. However, the specific pattern of these dispersals, and therefore the basis for the presence of A. sinensis and various fossil alligatoroids in Asia, remain unresolved. This problem is especially intriguing considering that none of the extant alligatoroids possesses a salt gland (Brochu, 1999), a part of the osmoregulatory system that excretes excess salt and whose presence therefore implies tolerance of high salinity. Partly because of their lack of salt glands, extant alligatoroids are largely restricted to freshwater environments, although some can inhabit brackish environments with easy access to freshwater. This was presumably also true of at least a subset of extinct alligatoroids, and would have limited their potential dispersal routes to ones that did not require crossing sea barriers.

The Asiatic alligatoroid fossil record is concentrated in the southeast part of the continent, and includes three extinct species and one unnamed specimen from China (Li & Wang, 1987; Young, 1964; Young, 1982; Skutschas et al., 2014), one extinct species from Thailand (Martin & Lauprasert, 2010), and possible Alligator sinensis remains from Thailand, Japan and Taiwan (Claude et al., 2011; Iijima, Takahashi & Kobayashi, 2016; Shan, Cheng & Wu, 2013). Although the Thai species Krabisuchus siamogallicus was recovered by Martin & Lauprasert (2010) as an alligatorine, and the unnamed specimen from the Maoming Basin of Guangdong Province, China was recovered by Skutschas et al. (2014) as an alligatorid of unresolved affinities, the three named Chinese extinct alligatoroid species have never been included in any cladistic analysis. Determining their phylogenetic positions would represent a significant step towards understanding alligatoroid evolution on a global scale.

The nominal Chinese fossil crocodylians referred to Alligatoroidea include Alligator luicus, Eoalligator chunyii and P. huiningensis (Li & Wang, 1987; Young, 1964; Young, 1982). A. luicus was established based on a single skeleton from the Miocene of Shandong Province (Li & Wang, 1987), E. chunyii based on material from the Paleocene of Nanxiong County, Guangdong Province (Young, 1964), and P. huiningensis based on a partial skull from the Paleocene of Huaining County, Anhui Province (Young, 1982). The two Eoalligator species pose problems at an alpha taxonomic level. Young (1964) assigned a number of undescribed and/or not even fully prepared bones to E. chunyii without comparing them explicitly to the holotype material, and his diagnosis of P. huiningensis (Young, 1982) did not include any comparisons with specimens of E. chunyii. Furthermore, Young (1964) erected not only E. chunyii but also a putative crocodyline species, Asiatosuchus nanlingensis. The specimens Young (1964) assigned to A. nanlingensis were similar in geographic and stratigraphic provenance to those he assigned to E. chunyii, but he did not compare the two species morphologically or identify any characters that distinguished them from one another. Surprisingly, A. nanlingensis shares strong morphological similarities with some of the specimens referred to E. chunyii, suggesting that the two species are synonymous even though they purportedly belong to different crocodylian lineages. Therefore, A. nanlingensis must be considered in any comprehensive revision of Eoalligator.

This paper re-evaluates the osteology, taxonomy and phylogenetic affinities of the single specimen of E. huiningensis and the more extensive material assigned to E. chunyii. The holotype and referred material of the crocodylid A. nanlingensis are also re-examined, and E. chunyii is shown to be a junior synonym of this species within Crocodylidae. As E. chunyii is the type species of Eoalligator, the new genus Protoalligator is established to accommodate the second species “Eoalligator” huiningensis, resulting in the new combination Protoalligator huiningensis. The results of this revision have important implications for future studies of alligatoroid evolution and paleobiogeography.

Materials

The numerous bones and bone fragments assigned by Young (1964) to Eoalligator chunyii were recovered from three localities that are all within 8 km of each other in Nanxiong County (Fig. 1): one southwest of Xiongzhou Town, and close to highway G 323 (locality 1, L1); one near Fengmenao Village, though its exact position is uncertain (L2); and one situated 1 km east of Xiuren Village (L3). L1 belongs to the lower Paleocene Shanghu Formation; L2 may be situated on the K-Pg boundary; and L3 belongs to the Upper Cretaceous Zhenshui Formation (Zhang et al., 2013). The specimens assigned to Asiatosuchus nanlingensis were recovered from L3 and two additional localities in the same county (Fig. 1): one 2 km northwest of Hukou village (L4), and one 4 km west of Xiongzhou Town (locality 5, L5). The specimens from L4 and L5 are from the Paleocene (Zhang et al., 2013). The material assigned by Young (1964) to E. chunyii includes six indeterminate specimens that cannot even be identified conclusively as crocodylian (and in some cases are clearly of chelonian origin), which are not considered further in this paper. Comprehensive information on the localities and stratigraphic positions of individual specimens is provided in Table 1 below.

Figure 1 Localities (L1–L5) in Nanxiong County, Guangdong Province that have produced specimens assigned to “Eoalligator chunyii” and/or Asiatosuchus nanlingensis.

Scale bar = 1 km.

Table 1 Material assigned by Young (1964) and Young (1982) to Eoalligator chunyii, Asiatosuchus nanlingensis and Eoalligator huiningensis, with detailed information of each specimen.

Specimen number	Identification	L	H	DS	ST	
Eoalligator chunyii	
Locality: Three localities in Nanxiong County, Guangdong Province (see also text and Fig. 1). Locality 1(L1), southeast of Xiongzhou Town; Locality 2 (L2), exact location uncertain, but near Fengmenao Village; Locality 3 (L3), 1 km east of Xiuren Village.	
Horizon: Zhenshui and Shanghu Formations, Cretaceous (?)—Paleocene.	
V2716-1.1	Posterior part of skull	L1	P	late	HT	
V2716-1.2	Incomplete right mandible	L1	P	late	HT	
V2716-2.1	Incomplete left premaxilla	L1	P	late	HT	
V2716-2.2	Incomplete left maxilla	L1	P	Early	HT	
V2716-2.3	Anterior part of left mandible	L1	P	Late	HT	
V2716-3	Left lower jaw in articulation with skull bones	L1	P	Late	HT	
V2716-4	Unidentifiable bones	L1	P	UN	HT	
V2716-5	Proximal end of left femur	L1	P	Early	HT	
V2716-6	Incomplete humerusa	L1	P	UN	HT	
V2716-7	Six disarticulated osteoderms	L1	P	Late	HT	
V2716-8	Unidentifiable bones	L1	P	UN	HT	
V2716-9	Unidentifiable bones	L1	P	UN	HT	
V2716-10	Unidentifiable bones	L1	P	UN	HT	
V2716-11	Incomplete skull	L1	P	Early	HT	
V2716-12	Incomplete right jugal with right ectopterygoid	L1	P	Late	HT	
V2716-13	Posterior part of the skull	L1	P	Early	HT	
V2716-14	Incomplete right jugal and dorsal part of postorbital	L1	P	Early	HT	
V2721.1	Left jaw	L3	K	Early	RS	
V2721.2	Left foota	L3	K	Early	RS	
V2721.3	Unidentifiable bone	L3	K	UN	RS	
V2721.4	A caudal vertebra	L3	K	Early	RS	
V2721.5	Incomplete right dentary	L3	K	Early	RS	
V2771	Incomplete left mandibleb	L2	P (?)	Early	RS	
Eoalligator huiningensis	
Locality: Huaidinghuawu Huaining, Anhui.	
Horizon: Wanghudun Formation, Paleocene.	
V4058	Anterior part of skull with lower jaw	HD	P	Early	HT	
Asiatosuchus nanlingensis	
Locality: Three localities in Nanxiong County, Guangdong Province (see also text and Fig. 1). Locality 3 (L3), 1 km east of Xiuren Village; Locality 4 (L4), 2 km northwest of Hukou Village; Locality 5 (L5), 4 km west of Xiongzhou Town; Locality 6 (L6), exact location uncertain, but near Fengmenao Village and L2.	
Horizon: Zhenshui and Shanghu Formations, Cretaceous (?)—Paleocene.	
V2721a	Fragment of mandible	L3	K	UN	RS	
V2772.1	Incomplete dentary	L5	P	UN	RS	
V2772.2	Incomplete dentary symphysisc	L5	P	UN	RS	
V2772.3	11 isolated teethc	L5	P	UN	RS	
V2772.4	Five isolated vertebrae	L5	P	UN	RS	
V2772.6	Three craniomandibular fragments	L5	P	UN	RS	
V2772.7	Incomplete right scapula	L5	P	UN	RS	
V2772.8	Incomplete left humerus	L5	P	UN	RS	
V2772.9	23 possible coprolites	L5	P	UN	RS	
V2773.1	Pair of incomplete mandibles	L4	P	Late	HT	
V2773.1a	Incomplete left dentary	L4	P	UN	RS	
V2773.2	Four isolated vertebrae	L4	P	Late	HT	
V2773.3	Left coracoid	L4	P	Late	HT	
V2773.4	Distal end of right femur	L4	P	Late	HT	
V2775	Posterior end of right mandibled	L6	P	Late	RS	
Notes.

Abbreviations DS Developmental Stage

H Horizon

HD Huaidinghuawu

HT Holotype

L Locality

RS Referred Specimen

ST Specimen Type

UN Uncertain

a non-crocodylian fossil.

b may not be conspecific with other material assigned by Young (1964) to E. chunyii.

c not conspecific with other material assigned by Young (1964) to A. nanlingensis.

d likely conspecific either with A. nanlingensis, or with unknown crocodilian represented by IVPP V 2772.2.

The nominal holotype of E. chunyii (IVPP V 2716) consists of a collection of bones from L1, some of which remained unprepared at the time of Young’s (1964) original description. The numbers IVPP V 2716-1 to IVPP V 2716-14 were assigned at some point to different parts of the holotype, although they were not used by Young, and recently additional numbers have been assigned to individual bones within IVPP V 2716-1 and V 2716-2. Young (1964) considered all the material included in IVPP V 2716 to “apparently” belong to one individual, but his description focused almost exclusively on a partial skull with the atlas-axis complex still attached (IVPP V 2716-1.1) and a fragment of a right mandible (IVPP V 2716-1.2). It is uncertain whether even these two specimens belong to a single individual. The additional material from L1 (see Table 1) is fragmentary, and includes a partial skull (IVPP V 2716-11) that clearly represents a different individual from IVPP V 2716-1.1. It is difficult to determine whether the other fragments from L1 belong to the individual represented by IVPP V 2716-1.1, the second individual represented by IVPP V 2716-11, or neither. The only other specimens ever referred to E. chunyii were additional ones listed in the original description (Young, 1964), including a single incomplete, poorly preserved left mandible (IVPP V 2771) from L2 and a few fragments from L3 (Table 1). The referred specimens, and even the components of the holotype other than IVPP V 2716-1, have never been fully described. In some cases (particularly that of IVPP V 2771), little evidence is available to support their conspecificity with IVPP V 2716-1.1, the most morphologically informative part of the holotype.

Young (1964) designated a collection of bones and coprolites from L4 (IVPP V 2773) as the holotype of A. nanlingensis, and referred additional mandibular and postcranial fragments from L3, L5 and L6 to the same species (Table 1). The most informative component of the holotype is a pair of large, incomplete mandibles (IVPP V 2773.1) that agree closely in size and morphology and probably come from the same individual. However, the holotype collection includes vertebrae of different sizes, as well as a fragment of a small mandible, so more than one individual is clearly represented in the material. Among the referred material is a collection from L5 that includes specimens probably referable to two separate taxa distinct from A. nanlingensis, as well as other fragments likely attributable to this species.

The only known specimen of Protoalligator (originally “Eoalligator”) huiningensis (IVPP V 4058), an incomplete skull with the lower jaw in place, was recovered from Huaidinghuawu in the southwest part of Huaining County, Anhui Province. The deposits that yielded this holotype specimen belong to the upper part of the Wanghudun Formation, and are thought to be middle Paleocene in age (Young, 1964).

For comparative purposes, several skeletons of extant crocodylians housed at the IVPP and at the University of the Witwatersrand, Johannesburg, South Africa were examined in connection with the present study. Additional data on the osteology of extant species were gleaned from photos and online data repositories.

Systematic paleontology

Crocodylia Gmelin, 1789, sensu Benton & Clark, 1988	
Globidonta Brochu, 1999	
Crocodyloidea Fitzinger, 1826	
Crocodylidae Laurenti, 1768	
Genus AsiatosuchusMook, 1940	
Asiatosuchus nanlingensisYoung, 1964	

Synonymy—Eoalligator chunyii Young, 1964

Revised diagnosis—Very large crocodylid with the following unique combination of characters (autapomorphy is indicated by an asterisk): well-developed sulcus present between supratemporal fenestrae (STF); two hemicondyles of quadrate subequal in size; short mandibular symphysis, extending to fourth dentary tooth; distinct sulcus present on dorsal part of lateral surface of surangular, adjacent to glenoid fossa*; surangular-articular suture anteroposteriorly oriented and situated within glenoid fossa; and proatlas shaped like isosceles triangle. Distinguished from other crocodylids by anteroposterior orientation of surangular-articular suture; from mekosuchines by similarity in size between quadrate hemicondyles; and from tomistomines and Asiatosuchus grangeri by shortness of mandibular symphysis, which extends only to level of fourth dentary tooth.

Description: material originally assigned to Eoalligator chunyii

Seven of the fragments that were originally assigned by Young (1964) to Eoalligator chunyii can be identified as adult based on size and fusion of neurocentral sutures (Brochu, 1996): IVPP V 2716-1.1, IVPP V 2716-1.2, IVPP V 2716-2.1, IVPP V 2716-2.2, IVPP V 2716-2.3, IVPP V 2716-3, IVPP V 2716-6, IVPP V 2716-12, and IVPP V 2721.4.

Figure 2 Asiatosuchus nanlingensis specimens originally assigned by Young (1964) to Eoalligator chunyii.

Including: IVPP V 2716-2.1, incomplete left premaxilla, in ventral (A) and lateral (B) views; IVPP V 2716-2.2, incomplete left maxilla, in dorsal (C), ventromedial (D) and posterior (E) views, a zoom in figure of a maxillary tooth is provided (D); IVPP V 2716-12, incomplete right jugal with ectopterygoid, in lateral (F) and ventral (G) views; IVPP V 2716-13, posterior part of skull, in dorsal (H) and occipital (I) views; IVPP V 2716-14, incomplete right jugal in lateral (J) and anterior (K) views, and dorsal part of right postorbital in dorsal view (L). Arrows point anteriorly. Abbreviations: bo, basioccipital; ect, ectopterygoid; exo, exoccipital; ITF, infratemporal fenestra; j, jugal; l, lacrimal; mx, maxilla; mxa, maxillary alveolus; mxt, maxillary tooth; nld, nasolacrimal duct; oc, occipital condyle; p, parietal; pm, premaxilla; pma, premaxillary alveolus; pmt, premaxillary tooth; so, supraoccipital; STF, supratemporal fenestra; sul, sulcus; vp, ventral process. Scale bars = 1 cm. Scale bars of the close-up on maxillary tooth (D) = 1 mm.

Skull

Premaxilla—IVPP V 2716-2.1 (Figs. 2A and 2B) comprises the lateral portion of a left premaxilla, with a small fragment of the maxilla still attached. In lateral view, the ventral margin curves posterodorsally as in Crocodylus porosus (Schneider, 1801). In Alligator sinensis the margin is less strongly curved. Four alveoli are preserved, and damage to the anterior end of the bone has probably removed one additional alveolus. Given that crocodylian premaxillae consistently contain either four or five teeth, the preserved alveoli are probably the second through fifth. Broken tooth crowns are preserved within the second, third and fifth alveoli. The third alveolus is the largest in diameter, the second and fourth considerably smaller and subequal in size, and the fifth alveolus is the smallest. Immediately posterior to the fifth alveolus, the premaxilla-maxilla suture runs laterally across a ventrally facing fossa. As in extant Alligator, this fossa accommodated the fourth dentary tooth, which would thus have been partly concealed when the jaw was closed. The first maxillary alveolus is situated just posterior to the recess, and is equal in diameter to the fifth premaxillary alveolus.

Maxilla—IVPP V 2716-2.2 is a partial left maxilla with a piece of the lacrimal attached, whereas the partial skull IVPP V 2716-11 includes large portions of both maxillae. In IVPP V 2716-2.2 (Figs. 2C–2E), only the posterior part of the left maxilla is preserved. This bone contains four alveoli with damaged teeth, rather than the five described by Young (1964). The second and fourth preserved teeth have relatively intact crowns, which are blunt with smooth (unserrated) carinae (Fig. 2D). The teeth are clearly somewhat laterally compressed, demonstrating that E. chunyii lacks the extremely bulbous, almost ball-shaped tooth crowns seen in typical globidontans.

Figure 3 Asiatosuchus nanlingensis specimen originally assigned by Young (1964) to Eoalligator chunyii.

Including: IVPP V 2716-11, incomplete skull, in dorsal (A, B) and ventral (C, D) views. Arrows point anteriorly. Abbreviations: ect, ectopterygoid; l, lacrimal; mx, maxilla; n, nasal; NAC, nasal channel; pal, palatine; pf, prefrontal; pm, premaxilla; pt, pterygoid; SOF, suborbital fenestra. Scale bars = 1 cm.

In IVPP V 2716-11 (Figs. 3A–3D), the dorsomedial portions of the two maxillae are preserved over a long interval representing the antorbital region of the skull. Each maxilla contacts the nasal medially and the lacrimal posteriorly. There is no maxillary process either intruding into the area occupied by the lacrimal or projecting between the lacrimal and nasal. Such a maxillary process is a derived character present in derived gavialoids, some Crocodylus species, most tomistomines and most alligatoroids (Brochu, 1999; Delfino & John, 2010; Langston, 1965). In alligatoroids, however, the location of the maxillary process is not always within the lacrimal. Poorly preserved sculpturing is visible on the dorsal surfaces of both maxillae. In ventral view, the nasal channel (NAC) is exposed anteriorly due to postmortem damage.

Lacrimal—Lacrimals are preserved in IVPP V 2716-2.2 and IVPP V 2716-11. In IVPP V 2716-2.2 (Figs. 2C–2E), the anterior part of the left lacrimal is present and contacts the maxilla anteriorly. In medial view, the nasolacrimal duct is apparent as a groove extending anteriorly towards the external naris (EN). The duct is “V”-shaped in cross section, with the apex pointing laterally.

In IVPP V 2716-11 (Figs. 3A and 3B), the anterior part of the left lacrimal is intact and contacts the maxilla anteriorly and the nasal medially. The posteromedial contact with the prefrontal may also be preserved, but the suture is not clearly apparent. Similarly, the right side of the skull retains part of the lacrimal and possibly a small part of the prefrontal, but sutural boundaries in this region are uncertain. However, the lacrimal certainly extends further anteriorly than the prefrontal, and contacts the nasal. Anterior extension of the lacrimal beyond the prefrontal is a plesiomorphic character, and the derived condition (shorter anterior extension of the lacrimal) is present in most alligatorines and nettosuchids (Brochu, 1999; Brochu, 2003).

Nasal—Both nasals are preserved in IVPP V 2716-11 (Figs. 3A–3D), although they are damaged anteriorly. Each nasal contacts its counterpart medially, the maxilla laterally, and the lacrimal posterolaterally, but the posterolateral contact with the prefrontal is not clearly preserved on either side of the skull. The lateral border of the nasal is straight, rather than concave as in A. sinensis.

Jugal—IVPP V 2716-12 comprises a fragment of a right jugal preserved with the ectopterygoid, and IVPP V 2716-14 includes a small piece of a right jugal. The portion of the bone preserved in IVPP V 2716-12 (Figs. 2E and 2F) appears to be a posterior fragment of the jugal, probably situated below the infratemporal fenestra (ITF). It is slightly disarticulated from the ectopterygoid. A recess with a sharp, shelf-like ventral margin is present on the lateral surface of the preserved part of the jugal. A recess in the equivalent position, approximately below the ITF, is absent in extant alligators and at least some Crocodylus species (C. porosus and C. niloticus) (Wang, pers. obs., 2016) (Laurenti, 1768). The lateral surface of the jugal is sculptured, and the contact with the quadratojugal is not preserved.

The jugal fragment in IVPP V 2716-14 (Figs. 2I and 2J) consists only of the base of the ascending process, which contributes to the postorbital bar, and the adjacent part of the jugal body. The preserved part of the postorbital bar is formed entirely by the jugal, implying that the descending process of the postorbital does not reach the base of the postorbital bar. There is no groove on the dorsal surface of the jugal between the base of the ascending process and the lateral margin of the bone.

Postorbital—IVPP V 2716-14 (Fig. 2K) includes a small postorbital that was originally attached by matrix to the partial right jugal. The two bones were separated during preparation, but the postorbital fragment reveals no important morphological information.

Ectopterygoid—A right ectopterygoid is preserved in IVPP V 2716-12, and both ectopterygoids are preserved in IVPP V 2716-11. In IVPP V 2716-12 (Figs. 2E and 2F), the body of the ectopterygoid is preserved but the maxillary and pterygoid processes are broken away. The anterior edge of the ectopterygoid, which forms the posterolateral margin of the suborbital fenestra (SOF), is strongly concave.

In IVPP V 2716-14 (Figs. 3A–3D), the medial part of the left ectopterygoid and a compressed portion of the right ectopterygoid are preserved. Each ectopterygoid contacts the maxilla laterally and the pterygoid medially. The narrow and deeply concave posterior apex of the left suborbital fenestra is enclosed by the pterygoid, but the ectopterygoid’s contribution to the margin of the fenestra begins only a short distance anterolateral to the apex.

Palatine—Both palatines are preserved in IVPP V 2716-11 (Figs. 3A–3D), but are damaged anteriorly. Each palatine contacts the pterygoid posteriorly. The posterior part of the lateral margin of the palatine runs posterolaterally. The palatine does not contribute to the concave posterior apex of the left SOF. The palatine-pterygoid suture extends transversely between the SOFs. Based on the position of the palatine-pterygoid suture, the choana must be surrounded entirely by the pterygoid, but the pterygoid is too poorly preserved for this to be confirmed by direct observation.

Pterygoid—The pterygoid is preserved in IVPP V 2716-1.1 and IVPP V 2716-11. The pterygoid of IVPP V 2716-1.1 (Figs. 4E and 4F) is represented only by a small fragment that remains in articulation with the braincase but has been deflected towards the right side of the skull. The fragment comprises the median part of the bone, and includes the small, damaged left and right posterior processes. The pterygoid is preserved in contact with the basioccipital dorsally and the quadrates dorsolaterally. The basisphenoid and laterosphenoid presumably also contact the preserved part of the pterygoid, but the boundaries of these bones are not clearly visible. Unfortunately, the orientation of the posterior process is uncertain due to distortion.

Figure 4 Asiatosuchus nanlingensis (IVPP V 2716-1.1), posterior part of skull originally designated by Young (1964) as part of the holotype of Eoalligator chunyii.

In dorsal (A, B), posterodorsal (C, D) and posteroventral (E, F) views. Abbreviations: AMP, attachment of adductor mandibulae posterior; at, atlas; ax, axis; bo, basioccipital; cf, carotid foramen; exo, exoccipital; f IX-X-XI, foramen for glossopharyngeal (IX), vagus (X) and accessory (XI) nerves; f XII, foramen for hypoglossal nerve; FO, foramen ovale; FM, foramen magnum; h, hypapophysis, nc, neural canal; o, osteoderm; OTF, orbitotemporal foramen; p, parietal; pr, proatlas; pt, pterygoid; PTC, opening of posttemporal canal; q, quadrate; qj, quadratojugal; s, squamosal; so, supraoccipital; STF, supratemporal fenestra; sul, sulcus. Scale bars = 1 cm.

In IVPP 2716-11 (Figs. 3C and 3D), the anteromedial part of the pterygoid is preserved but the posterior part is severely damaged. The pterygoid contacts the palatine anteriorly and the ectopterygoid anterolaterally.

Quadratojugal—The posterolateral part of the right quadratojugal is preserved in IVPP V 2716-1.1 (Figs. 4A–4D, 5A), as a small piece of bone attached to the lateral margin of the quadrate.

Figure 5 Asiatosuchus nanlingensis (IVPP V 2716-1.1), posterior part of skull originally designated by Young (1964) as part of the holotype of Eoalligator chunyii.

In right lateral (A), right dorsoanterior (B) and left lateral (C) view. Abbreviations: ax, axis; cr, cervical rib; h, hypapophysis; o, osteoderm; or, otic recess; p, parietal; q, quadrate; qj, quadratojugal; s, squamosal; vt, ventral tuberosity. Scale bars = 1 cm.

IVPP V 2716-3 comprises the posterior end of a left mandible preserved in contact with the quadrate region of the skull. The posterior part of the quadratojugal is preserved, and contacts the quadrate medially (Figs. 6A and 6B). The quadratojugal extends to the posterior tip of the lateral hemicondyle of the quadrate, obscuring the hemicondyle in lateral view. Anteriorly, the root of the quadratojugal spine (QJS) remains intact (6A). The position of this structure suggests the spine would have protruded out into the infratemporal fenestra rather than following the superior margin of this opening, a plesiomorphic character shared with Gavialis gangeticus, Crocodylus and Osteolaemus. A mediodorsally placed quadratojugal spine is a derived feature shared by Trilophosuchus rackhami (Willis, 1993) and Crocodylus depressifrons (Delfino & Smith, 2009), and most alligatorines (Brochu, 1999; Brochu, 2011). Sculpturing is preserved on the lateral surface.

Figure 6 Asiatosuchus nanlingensisspecimens originally assigned by Young (1964) to Eoalligator chunyii.

Including: IVPP V 2716-3, left jaw articulated with part of skull, in dorsal (A) and lateral (B) views; IVPP V 2716-1.2 (part of the holotype of E. chunyii), two incomplete right mandibles, in lateral (C) and medial (D) views; IVPP V 2716-2.1, left dentary, in dorsal (E) and medial (F) views; IVPP V 2721.5, anterior part of right dentary, in dorsal (G) and ventral (H) views; IVPP V 2771, left mandible, in dorsal (I), lateral (J) and medial (K) views; IVPP V 2721.1, left posterior mandibular fragment in dorsal (L) and lateral (M) views. Most specimens are referable to Asiatosuchus nanlingensis, but the taxonomic identity of IVPP V 2771 is uncertain. Arrows point anteriorly. Abbreviations: an, angular; art, articular; CQC, cranioquadrate canal; d, dentary; da, dentary alveolus; dt, dentary tooth; exo, exoccipital; FA, foramen aerum; fw, full width; mg, Meckelian groove; o, osteoderm; q, quadrate; qj, quadratojugal; qjs, quadratojugal spine; sa, surangular; sp, splenial; sul, sulcus; sy, symphysis; tpb, tortoise pleural bone. Scale bars = 1 cm.

Quadrate—One or both quadrates are preserved in IVPP V 2716-1.1, IVPP V 2716-3 and IVPP V 2716-13. In IVPP V 2716-1.1 (Figs. 4A–4D, 5A and 5B), the right quadrate is relatively complete apart from damage to the margins and the absence of the condyles, but the left quadrate is less well-preserved. The quadrate contacts the squamosal dorsally, the parietal and the exoccipital medially, the pterygoid ventrally, the quadratojugal laterally and the laterosphenoid anteriorly. Anteriorly, the quadrate and laterosphenoid surround the dorsal part of the foramen ovale (FO), meeting at a nearly vertical suture that intersects the dorsal border of the FO. In the midline of the skull, a remnant of the basisphenoid contributes to the floor of the braincase and is visible in anterior view because the front part of the skull is missing. Below the posterior rim of the supratemporal fenestra (STF), the quadrate appears to contribute to the floor of the orbitotemporal foramen (OTF) (Figs. 4A, 4B and 5B), intruding between the parietal and squamosal. This character is shared with L.canadensis, Brachychampsa and Deinosuchus (Brochu, 1997; Gilmore, 1911; Norell, Clark & Hutchison, 1994). In the posteroventral view, there is a crest on the ventral side of quadrate for attachment of the adductor mandibulae posterior (AMP). An equivalent, but less well-developed, crest is present in A. sinensis.

In IVPP V 2716-3 (Fig. 6A) the body of the left quadrate is preserved, and contacts the exoccipital dorsomedially and the quadratojugal laterally. The lateral hemicondyle of the quadrate is covered by the quadratojugal laterally. As in other crocodylians, there is no thin crest on the dorsal surface of the quadrate. The medial hemicondyle of the quadrate is larger than the lateral hemicondyle, as in extant Crocodylus. No foramen aerum (FA) is evident on the dorsal surface. The dorsal surface above the medial hemicondyle is damaged, as is the nearby part of the medial side of the quadrate. Furthermore, the ventromedial corner of the quadrate has broken away from the rest of the bone along a vertical crack and has been displaced a short distance posteriorly. The FA may have originally been located either on the line of the crack or on the medial surface of the quadrate. The latter possibility seems more likely, given that the FA on the articular is located close to the medial edge of the dorsal surface (see below). A fragment of the paroccipital process is preserved in articulation with the quadrate. Ventromedial to this fragment, the cranioquadrate canal (CQC) is exposed by post mortem damage.

Fragments of the ventral process of the left quadrate are preserved in the damaged partial skull IVPP V 2716-13, and the crest for the AMP is visible as in IVPP V 2716-1.1. However, no other important morphological details are discernable.

Laterosphenoid—A remnant of the laterosphenoid is preserved in IVPP V 2716-1.1, near the foramen ovale (FO). It contacts the quadrate posteriorly, the parietal dorsally and the pterygoid ventrally, although the suture with the pterygoid is unclear.

Parietal—The parietal is preserved in IVPP V 2716-1.1 and IVPP V 2716-13. In IVPP V 2716-1.1 (Figs. 4A–4D, 5B), the parietal is nearly complete, except that the anterior part of the bone is damaged. The parietal contacts the squamosals laterally, the exoccipitals posterolaterally, and the supraoccipital posteriorly, and exhibits an inverted “Y” shape in dorsal view. The intertemporal bar bears a deep longitudinal sulcus that is absent in extant Alligator, and less distinct in adult Crocodylus niloticus than in IVPP V 2716-1.1. The rims of the STFs are elevated above the skull table. A similar, but less distinct, rim is present in “Crocodylus” depressifrons (Delfino & Smith, 2009). The sculpturing on the surface of the parietal is better-developed near the squamosal than elsewhere.

In IVPP V 2716-13 (Fig. 2H) the middle of the posterior portion of the parietal is preserved, but the surface is badly damaged. The parietal has also been displaced to the left relative to the occipital condyle and the rest of the braincase. The sulcus between the STFs is less well-developed than in IVPP V 2716-1.1, and the intertemporal bar is substantially broader. Both differences might reflect ontogenetic variation, as IVPP V 2716-13 is considerably smaller than IVPP V 2716-1.1.

Squamosal—The medial part of the left squamosal, and most of the right squamosal apart from the anterior process, are preserved in IVPP V 2716-1.1 (Figs. 4A–4D, 5A and 5B). The squamosal contacts the parietal medially, the quadrate ventrally, the supraoccipital posteromedially, and the exoccipital ventrally. The posterior process extends posterolaterally, rather than posteriorly as in extant alligators, and reaches the posterior tip of the exoccipital process. In occipital view, the squamosal-exoccipital suture extends horizontally, but its lateral portion has an irregular ventral concavity. In lateral view, the squamosal-quadrate suture runs alongside the posterior margin of the otic recess for a short distance before curving anteriorly to intersect the edge of the recess. The posterior margin of the otic recess forms a small posterior notch (Fig. 5A).

Supraoccipital—The supraoccipital is completely preserved in IVPP V 2716-1.1 and IVPP V 2716-13. The supraoccipital contacts the parietal anteriorly, the squamosal laterally and the exoccipital ventrolaterally. In IVPP V 2716-1.1 (Fig. 4), the portion of the supraoccipital that is exposed on the skull table is semicircular. The supraoccipital is excluded from the foramen magnum (FM) by both exoccipitals.

In IVPP V 2716-13 (Figs. 2G and 2H), the supraoccipital is less well-preserved and well-exposed. The supraoccipital appears morphologically similar to its counterpart in IVPP V 2716-1.1, but makes a smaller contribution to the skull table.

Exoccipital—One or both exoccipitals are preserved in IVPP V 2716-1.1, IVPP V 2716-3 and IVPP V 2716-13. In IVPP V 2716-1.1 (Fig. 4) the entire right exoccipital and the medial part of the left exoccipital are present. Each exoccipital contacts the supraoccipital dorsomedially, the squamosal dorsally, the quadrate ventrolaterally and the basioccipital ventromedially. The exoccipital forms the lateral margin of the foramen magnum (FM). Further laterally, the exoccipital bears a tapering ventral flange that extends to the same level as the ventral margin of occipital condyle. Two foramina are preserved lateral to the occipital condyle and medial to the suture with the quadrate: the dorsolaterally positioned jugular foramen transmitting the glossopharyngeal (IX), vagus (X), and accessory (XI) nerves, and the medioventrally positioned carotid foramen. Situated further dorsally, and lateral to the FM, is the hypoglossal canal for branches of the hypoglossal nerve (XII).

Only a small fragment of the exoccipital, preserved in contact with the quadrate, is present in IVPP V 2716-3. Similarly, only fragments of the right exoccipital are preserved in IVPP V 2716-13 (Figs. 2G and 2H). The sutural boundaries of the bone are unclear due to postmortem damage. However, the exoccipital has a ventral flange of about the same extent seen in IVPP V 2716-1.1.

Basioccipital—The basioccipital is complete in IVPP V 2716-1.1 and IVPP V 2716-13. In IVPP V 2716-1.1 (Figs. 4 and 5C) the basioccipital contacts the exoccipitals dorsolaterally and the pterygoid ventrally, and contributes to the ventral part of the FM. The occipital condyle is wide and suboval in occipital view, being somewhat dorsoventrally compressed, and projects posteriorly. A small basioccipital tuber appears to be present on the left side of the basioccipital, but there is no evidence of the midline crest below the occipital condyle that combines with the tubera to define a “W” shape in ventroccipital view (Fig. 4E) in extant alligators and C. niloticus.

Lower jaw

Dentary—In the partial mandible IVPP V 2716-1.2 (Figs. 6C and 6D), the middle part of the right dentary is in contact with the medially positioned splenial, and the entire preserved section of the mandible has been laterally compressed. A second mandibular fragment has been attached with plaster to the posterior end of the main fragment, and was tentatively described by Young (1964) as though this represented its natural position. However, the second fragment appears to represent a small portion of the opposite (left) mandible of either the same individual or another of similar size.

Anteriorly, the main fragment of IVPP V 2716-1.2 is preserved almost to the level of the symphysis. The upper margin of the dentary as seen in lateral view has a slight undulating curvature in most crocodylians, although this margin is straight in some taxa such as Gavialis gangeticus. In some alligatoroids the upper margin of the dentary is more strongly curved, and in these cases the lower margin tends to become convex. In IVPP V 2716-1.2 the lower margin of the preserved part of the dentary is straight, whereas the dorsal margin is relatively straight posteriorly, rises to a low eminence containing sockets for two relatively large teeth, and slopes gently anteroventrally towards the anterior end of the bone. The curvature of the mandible thus appears to match the gently curved condition of typical crocodylians.

The preserved series of alveoli extends along the entire preserved part of the main fragment and may represent the fourth to seventeenth dentary teeth, but the anteriormost and posteriormost alveoli are incomplete. Young’s (1964) estimate that twenty-three teeth would have been present in life included the alveoli present on the posterior fragment. The incomplete anteriormost alveolus on the main fragment, which we identify as the fourth, appears to be the largest in the series. The fifth through ninth alveoli are relatively small, whereas the tenth and particularly the eleventh are somewhat enlarged. The twelfth alveolus is considerably smaller than the eleventh. The more posterior alveoli are also relatively small, and appear to have suffered from lateral compression.

The posterior fragment included in IVPP V 2716-1.2 contains five alveoli that probably represent teeth 10-14 of a left mandible, based on their size and the presence of an anteriorly widening shelf medial to the tooth row. This fragment corresponds closely in morphology to the equivalent part of the main fragment. The medial surface of the posterior fragment bears a subdued area, bounded dorsally by the prominent medial edge of the shelf, which may represent part of the articular surface for the splenial. Below this is a longitudinal furrow that presumably corresponds in part to the Meckelian groove, but seems to have been enlarged by damage to the mandible.

IVPP V 2716-2.3 is the anterior part of a left dentary (Figs. 6E and 6F), whereas IVPP V 2721.5 is the anterior part of a right dentary (Figs. 6G and 6H). IVPP V 2771 is a poorly preserved left dentary that may not pertain to the same species as the other material (Figs. 6I–6K), because it shows no diagnostic characters of A. nanlingensis and is the only crocodylian specimen recovered from L2. An anteriorly tapering depression on the medial surface of IVPP V 2716-2.3 represents the anterior end of the Meckelian groove. The mandibular symphysis consistently extends posteriorly to the level of the fourth dentary tooth, but without reaching the tooth’s posterior margin. Five alveoli (for teeth 4–8) are preserved in IVPP 2716-2.3, whereas four alveoli (for teeth 4–7) are preserved in IVPP V 2721.5 and eight indeterminate alveoli are preserved in IVPP V 2771. The part of IVPP V 2771 immediately posterior to the symphysis is relatively intact, and may include the splenial, but more posteriorly the mandible is damaged, distorted and laterally compressed. The fourth tooth is large, although the apex is missing. About 2 cm posterior to the fourth tooth is preserved a small tooth with an acute, laterally compressed crown and smooth carinae (Fig. 6J). A tortoise pleural bone (TPB) is attached to the medial surface of the mandible, near the posterior end of the preserved portion.

Splenial—The main fragment of IVPP V 2716-1.2 (Figs. 6C and 6D) includes a preserved partial splenial, whose anterior tip is slightly broken but clearly would have closely approached the symphysis in the intact mandible (Figs. 6E and 6F). The splenial contacts the dentary laterally, and the medial surface of the former bone lacks any evident perforation. In IVPP V 2771 a fragment of the splenial may be present immediately caudal to the symphysis (Figs. 6I–6K). In IVPP V 2716-2.3, the Meckelian groove ends a short distance posterior to the mandibular symphysis, implying the splenial would not quite have entered the symphysis even if it covered the groove completely.

Surangular—The left surangular is preserved in both IVPP V 2716-3 and IVPP V 2721.1. In IVPP V 2716-3 (Figs. 6A and 6B) the surangular is nearly complete, although the ventral part of the bone is distorted. The surangular contacts the articular medially and the angular posteroventrally. The dorsal margin of the surangular is flush with the transverse ridge at the posterior edge of the glenoid fossa, so that the surangular fully conceals the glenoid fossa in lateral view. Immediately lateral to the glenoid fossa, there is a sulcus on the narrow dorsolateral margin of the surangular. This sulcus is a unique feature among crocodylians, seen only in specimens originally assigned to Eoalligator chunyii and Asiatosuchus nanlingensis, although a smaller, more posteriorly positioned sulcus is present on the surangular of the European alligatoroid Diplocynodon (Figs. 5 and 8 in Martin et al., 2014). The surangular extends posterodorsally, covering the anterior part of the lateral surface of the retroarticular process. However, it is uncertain whether the surangular reaches the distal end of the retroarticular process, which is damaged.

In IVPP V 2721.1 (Figs. 6L and 6M) the sulcus next to the glenoid fossa is better developed than in the considerably larger IVPP V 2716-3, which indicates that the sulcus becomes less distinct during ontogeny. The surangular-articular suture runs anteroposteriorly near the lateral edge of the glenoid fossa. Participation of the surangular in the glenoid fossa is feature shared with crocodyloids.

Angular—The left angular is preserved in IVPP V 2716-3 and IVPP V 2721.1 (Figs. 6A, 6B and 6M). The angular contacts the surangular anterodorsally and the articular medially, extending up the retroarticular process.

Articular—The left articular is preserved in IVPP V 2716-3 and IVPP V 2721.1. In IVPP V 2716-3 (Fig. 6B) the articular is nearly complete, although the posterodorsal tip of the retroarticular process is missing. This bone is covered laterally by the angular and surangular. In medial view, no lingual foramen is evident either on or near the articular-surangular suture. In contrast to the condition in many crocodylians, the articular does not form a lamina that projects forward to form an overlapping contact with the medial face of the surangular. In dorsal view, the foramen aerum (FA) is visible on the medial part of the articular. The presence of an FA on the articular is typical among crocodylians, but in alligatoroids the FA is displaced dorsolaterally, rather than remaining medially positioned as in IVPP V 2716-3 (Brochu, 1999).

In IVPP V 2721.1 (Figs. 6L and 6M), the left articular is nearly intact, although the posterodorsal tip is damaged. The articular is smaller than that of IVPP V 2716-1. The glenoid fossa contains a small longitudinal ridge separating the recesses that accommodate the two hemicondyles of the quadrate. The medial recess is much narrower than the lateral one.

Postcrania

Proatlas—A proatlas is preserved in IVPP V 2716-1.1 (Figs. 4A–4D and 5C), and is complete but displaced from its original position. The proatlas approximates the shape of an isosceles triangle in dorsal view. Anteriorly, it forms an obtuse point, similar to the apex of the proatlas in Alligator sinensis (Cong et al., 1998) but quite different from the prominent anterior process characterizing Diplocynodontinae (Brochu, 1999). The sagittal crest is damaged, but appears to extend anteroposteriorly. A small ventral tuberosity (Fig. 5C), representing the point of attachment of the atlantoccipital ligament (Cong et al., 1998), is situated halfway along the exposed left anterolateral surface of the proatlas. The posterior margin of the proatlas is linear instead of smoothly concave as in Crocodylus acutus (Cuvier, 1807).

Atlas—The left half of the neural arch of the atlas is preserved, surrounded by matrix, in IVPP V 2716-1.1 (Figs. 4A–4D). An embayment is present in the anterior margin of the neural arch, and the proatlas likely articulated with a tapering anterior prominence that is situated dorsal to the embayment. The atlas intercentrum is preserved separately, positioned to the right of the occipital condyle and close to the axis. The intercentrum is similar to the proatlas in size, and has a transversely broad ventral surface that would be tilted to face somewhat anteriorly if the skull and cervical vertebrae were in natural articulation (anterior surface of Brochu, 1999, Fig. 39). In contrast to the condition in extant Alligator (Brochu, 1999), the posterior margin of the ventral surface does not form a recess between the two posterolaterally facing facets for the atlantal ribs, and the intercentrum as a whole is not anteroposteriorly compressed.

Axis—The axis is almost completely preserved in IVPP V 2716-1.1 with the odontoid process (atlantal centrum) in place (Figs. 4A–4D), although only the base of the neural spine is present. The odontoid process is not fused to the axis centrum, and remains in contact with the atlantal intercentrum although this joint has undergone slight disarticulation. On the anterior part of the lateral surface of the centrum is a small longitudinal crest, dorsal and ventral to which are two shallow recesses. The hypapophysis on the ventral surface of the centrum is long and well-developed, occupying the anterior half of the centrum’s length. What appears to be the first two right cervical ribs are preserved in contact with the right side of the axis, although not in their natural positions (Fig. 5C). Further anteriorly, a displaced rib from one of the postaxial cervicals is preserved adjacent to the atlantal intercentrum and the left quadrate.

Caudal Vertebra—IVPP V 2721.4 is a disarticulated, procoelous caudal vertebra (Figs. 7A–7C). The prezygapophysis extends further laterally than the postzygapophysis. Remnants of the transverse processes indicate that these structures were originally wide and well-developed, extending for half the length of the centrum. A ventral groove (VG) is present on the ventral surface of the centrum near the posterior condyle. The presence of the VG suggests that this vertebra is from the anterior part of the tail. There is no visible neurocentral suture, indicating that this specimen probably represents a mature individual. However, it could still be a sub-adult, as the sutures close first in the caudal vertebrae during ontogeny (Brochu, 1997).

Figure 7 Asiatosuchus nanlingensis specimens originally assigned by Young (1964) to Eoalligator chunyii.

Including: IVPP V 2721.4, caudal vertebra, in dorsal (A) ventral (B) and lateral (C) views; IVPP V 2716-5, proximal end of left femur, in lateral (D) and medial (E) views; IVPP V 2716-7, six disarticulated osteoderms, in dorsal (F) and ventral (G) views. Abbreviations: ns, neural spine; prz, prezagapophysis; tp, transverse process; vg, ventral groove. Scale bars = 1 cm.

Femur—IVPP V 2716-5 is a left femoral head (Figs. 7D and 7E), resembling in gross morphology the equivalent structure in extant crocodylians. Judging from its size, the femur belongs to an immature individual.

Osteoderms—IVPP V 2716-7 consists of a cluster of six disarticulated osteoderms with dorsal keels (Figs. 7F and 7G). Two of the osteoderms are distinctly narrower in the direction perpendicular to the keel, whereas three are approximately square. The sixth osteoderm is incomplete and concealed by the others. The narrow osteoderms have keels that extend for almost their entire lengths, whereas the keel of the only well-preserved and well-exposed square osteoderm is separated from the embayed anterior margin by a small subdued area. The pits along the anterior margin of this osteoderm are small and transversely narrow in the vicinity of the keel, whereas the other visible pits are larger and more round. The sculpturing of the narrow osteoderms appears to consist entirely of round pits.

Figure 8 Specimens assigned by Young (1964) to Asiatosuchus nanlingensis.

Including: IVPP V 2773 (holotype), left mandible in dorsal (A), lateral (B) and medial (C) views, and right mandible in lateral (D) and medial (E) views; An incomplete fragment of dentary possibly the anterior part of the holotype mandible in lateral (F) and medial (G) view; IVPP V 2772.1, right dentary in lateral (H) and medial (I) views; IVPP V 2772.2, dentary symphysis in dorsal (J) and ventral (K) views; IVPP V 2721a, dentary fragment in lateral (L) and medial (M) views. IVPP V 2772.2 appears to represent a longirostral crocodylian distinct from A. nanlingensis. Abbreviations: da, dentary alveolus; dt, dentary tooth; EMF, external mandibular fenestra. Scale bars = 2 cm.

Description: material originally assigned to Asiatosuchus nanlingensis

The most informative component of the holotype of Asiatosuchus nanlingensis (IVPP V 2773) is a pair of very large incomplete mandibles (IVPP V 2773.1), each representing the articular region and a part of the ramus (Figs. 8A–8E). A small section of dentary containing four alveoli, with the corresponding piece of the splenial still in place medially (IVPP V 2773.5; Figs. 8F and 8G), may represent an anterior fragment of the right mandible of IVPP V 2773.1 as suggested by Young (1964). The size and proportions of the fragment seem compatible with those of the large mandibles, under the assumption that the fragment originally occupied an anterior position within the right mandible, and the alveoli in the isolated fragment are similar in diameter to those preserved in the left mandible. The dorsal margin of the fragment slopes gradually anteroventrally, a feature consistent with an anterior placement within the lower jaw. The ventral part of the external surface of the fragment is heavily sculptured, but the fragment provides little morphological information.

The left mandible of IVPP V 2773.1 preserves three incomplete teeth, lacking the crowns. There are two empty alveoli immediately posterior to the teeth, and a third immediately anterior to them. A single displaced tooth, bearing smooth carinae, is preserved in contact with the medial surface of the right mandible. Young (1964) considered the small size of the external mandibular fenestra (EMF) in both mandibles to be a diagnostic character, but this condition appears exaggerated by dorsoventral compression. The curvature of the dorsal and ventral margins of the EMF suggests, in both jaws, that this structure was originally larger. In particular, the ventral margin is distinctly concave, implying that the EMF was both larger and rounder than the slit-shaped opening seen in Planocrania hengdongensis (Li, 1984). However, it is difficult to be sure whether the intact EMF would have been very large, as in Alligator, or only moderately large as in Crocodylus niloticus. Based on this interpretation, the EMF is similar to that of Asiatosuchus grangeri (Mook, 1940). Most sutures on both mandibles are fused, presumably as a result of aging. There is no foramen aerum (FA) preserved on either articular. It is conceivable that the FA closed as the individual continued to develop ontogenetically, but there is no evidence to support closure of the FA with advancing age in any crocodylian. On the dorsolateral part of the surangular, IVPP V 2773.1 has a similar sulcus seen in IVPP V 2716-3 (Figs. 8A and 8B), although this sulcus is smaller in size. Inferring from the relative sizes between IVPP V 2773.1 and IVPP V 2716-3, the smaller sulcus in IVPP V 2773.1 may again reflect an ontogenetic variation, where the surangular sulcus degenerates as the crocodylian ages. Such sulcus is highly diagnostic at least within Crocodyloidea, which reflects the existence of taxonomic problems between A. nanlingensis and E. chunyii.

As originally designated by Young (1964), the holotype also includes four cervical and dorsal vertebrae (IVPP V 2773.2), an incomplete left coracoid (IVPP V 2773.3), and the distal end of a right femur (IVPP V 2773.4). Two of the vertebrae are much smaller than the others, and clearly do not belong to the same individual represented by either of the large vertebrae or by the large mandibles. All four vertebrae are procoelous, as expected in any eusuchian.

Two mandibular fragments (IVPP V 2772.1 and IVPP V 2772.2) from a site (L5) several km from the locality that produced the holotype (L4) were both referred to A. nanlingensis by Young (1964), but differ from one another in some important morphological features. The articular region is absent in both cases, limiting the potential for comparisons with the holotype mandibles. In IVPP V 2772.1 (Figs. 8H and 8I), the mandibular symphysis extends to what is almost certainly the fourth dentary tooth, based on its large diameter. This corresponds to the condition in specimens assigned to Eoalligator chunyii (IVPP V 2716-1.2), but distinguishes IVPP V 2772.1 from Asiatosuchus grangeri from the Eocene of Inner Mongolia, in which the symphysis extends back to the sixth dentary tooth (Mook, 1940). In IVPP V 2772.2, however, the symphysis extends to an alveolus that may be interpreted as the seventh, if the first large alveolus is identified as the fourth. Also, IVPP V 2772.2 (Figs. 8J and 8K) has a dorsoventrally compressed symphysis, unlike IVPP V 2772.1 and specimens assigned to Eoalligator chunyii but as in longirostral taxa such as Gavialis gangeticus (Gmelin, 1789). In IVPP V 2772.2, the fourth dentary alveolus is smaller than all three posterior alveoli seen in IVPP V 2773, indicating that IVPP V 2772.2 must represent a smaller individual. Based on the length and dorsoventral compression of the mandibular symphysis, IVPP V 2772.2 likely represents a different species from the holotype, in which the middle part of the left mandibular ramus is sufficiently deep to indicate that the skull is not particularly longirostral. IVPP V 2772.1 is similar to IVPP V 2716-1.2 in that the dentary symphysis extends back to the fourth dentary alveolus, and in that the tenth to twelfth dentary alveoli are enlarged. These morphological similarities and the provenance of IVPP V 2772.1 suggest that referral of this specimen to A. nanlingensis is likely valid, although some uncertainty remains given that IVPP V 2772.1 shares no highly distinctive synapomorphies with either the holotype of A. nanlingensis or specimens referred to E. chunyii. IVPP V 2772.2, by contrast, seems to represent an unidentified longirostral crocodylian different from A. nanlingensis.

IVPP V 2772 also includes 11 isolated teeth (V 2772.3), a number of coprolites and possible coprolites (V 2772.9), and several additional non-diagnostic cranial and postcranial fragments (Table 1). All of the material, except possibly some of the coprolites, does appear to be of crocodyliform origin, but the isolated teeth have serrated carinae and therefore represent a taxon distinct from A. nanlingensis. Their conical morphology implies that they are unlikely to belong to either species of Planocrania (Li, 1976; Li, 1984), as Planocrania has laterally compressed teeth. They are also unlikely to be conspecific with the longirostral form represented by IVPP V 2772.2, because no Paleocene longirostral crocodyliform is known to have a ziphodont dentition. The teeth are not necessarily from a crocodylian, but it is notable that five vertebrae included in the holotype collection (IVPP V 2772.4) are procoelous, implying that they are at least of eusuchian origin. These vertebrae, along with the other indeterminate cranial and postcranial fragments in the sample, could belong to any of the three species (the longirostral form, the ziphodont form and tentatively identified A. nanlingensis) present at L5.

IVPP V 2721a (Figs. 8L and 8M) is a small dentary fragment from a locality that has also produced specimens assigned to Eoalligator chunyii (L3). This specimen was listed by Young (1964) as belonging to A. nanlingensis. The fragment clearly belongs to a large individual, and bears some sculpturing on the lateral surface, but no diagnostic morphological features are preserved. Its provenance suggests it likely belongs to A. nanlingensis, assuming E. chunyii is indeed a junior synonym of this taxon.

IVPP V 2775, the only specimen in this study recovered from L6, is the posterior end of a large right crocodylian mandible referred to A. nanlingensis by Young (1964). This poorly preserved fragment shows no diagnostic features, and could conceivably belong to either A. nanlingensis or the unidentified longirostral taxon represented by IVPP V 2772.2.

Systematic paleontology

Crocodylia Gmelin, 1789, sensu Benton and Clark, 1988	
Globidonta Brochu, 1999	
Alligatoroidea Gray, 1844, sensu Brochu, 2003	
Protoalligator, gen. nov.	

Type species—Protoalligator huiningensis (Young, 1982)

Synonym—Eoalligator huiningensis Young, 1982

Etymology—From Greek protos, “first”, combined with the name of the extant genus Alligator. This coinage retains the spirit of Young’s original but now invalid genus name Eoalligator (Greek eos, “dawn” or “primordial”, combined with Alligator).

Horizon and Locality—Wanghudun Formation, middle Paleocene. Huaidinghuawu, Anhui Province, China (Young, 1982).

Revised diagnosis—A short snouted alligatoroid with the following unique combination of characters: posterior process of premaxilla protruding into external naris; pit for fourth dentary tooth present between premaxilla and maxilla; eleventh and twelfth dentary teeth largest in mandible posterior to fifth dentary tooth. Distinguished from all other alligatoroids outside Alligator by protrusion of posterior process of premaxilla into external naris; from most alligatoroids by presence of pit for fourth dentary tooth between premaxilla and maxilla; and from alligatorines by large size of eleventh and twelfth dentary teeth.

Description

Protoalligator huiningensis was originally named Eoalligator huiningensis, the second nominal species of Eoalligator (Young, 1982). The holotype and only known specimen comprises a snout and the anterior parts of both mandibles (Figs. 9 and 10). Despite the fact that P. huiningensis and some specimens that we refer to Asiatosuchus nanlingensis were originally assigned to the same genus, few taxonomically significant features can be compared between the two species. There are two evident differences: (1) the apex of the fourth dentary tooth in P. huiningensis is occluded by a pit between the premaxilla and maxilla, while no such pit exists in A. nanlingensis; and (2) the mandibular symphysis of P. huiningensis extends to the fifth dentary tooth, while that of A. nanlingensis extends only to the fourth. Conversely, only limited and taxonomically unimportant morphological similarities exist between these two species. The following description of P. huiningensis is intended to supplement that of Young (1982).

Figure 9 Holotype of Protoalligator huiningensis (IVPP V 4058).

Anterior part of skull and lower jaws in dorsal (A, B), ventral (C, D) views. Abbreviations: an, angular; c, coracoid; d, dentary; dp, depression; EN, external naris; EMF, external mandibular fenestra; FIM, foramen intermandibularis medialis; j, jugal; l, lacrimal; mx, maxilla; n, nasal; pm, premaxilla; sp, splenial. Scale bars = 2 cm.

Figure 10 Holotype of Protoalligator huiningensis (IVPP V 4058).

Anterior part of skull and lower jaws in right dorsolateral (A, B), left ventrolateral (C, D) views and the anteroventral view of the premaxilla (E). Abbreviations: an, angular; c, coracoid; d, dentary; dp, depression; EN, external naris; EMF, external mandibular fenestra; FIM, foramen intermandibularis medialis; j, jugal; l, lacrimal; mx, maxilla; n, nasal; pm, premaxilla; sa, surangular; sp, splenial. Scale bars = 2 cm.

Skull

Premaxilla—The premaxillae (Figs. 9 and 10) form the anterolateral margins of the naris, which appears oval in dorsal view. Each premaxilla contacts its counterpart medially and the maxilla posteriorly. The premaxilla bears four slender teeth (Fig. 10E), which increase in length posteriorly. The fourth is the largest in diameter, as in extant alligators and Crocodylus niloticus, whereas in Asiatosuchus nanlingensis the third premaxillary tooth is largest. Near the contact with the maxilla, the lateral margin of the premaxilla is recessed to form the anterior part of the pit for the fourth dentary tooth, which is exaggerated by compression. At the anterior margin of the external naris (EN), the two premaxillae combine to form a posterior process that protrudes a short distance into the EN but is clearly incomplete. In basal Alligatorinae, there is no such process, and the nasal bar is formed entirely by the nasals and fails to fully bisect the EN. In Alligator, which is characterized by a complete nasal bar that bisects the EN (Wermuth, 1953; Malone, 1980; Norell, 1988; Brochu, 1999), the anterior part of the bar is always formed by a process of the premaxillae. The presence of a posterior process of the premaxillae may therefore indicate that the nasal bar completely bisects the EN in the intact skull of P. huiningensis, although confirmation of this possibility will require better-preserved material.

Maxilla—The maxillae (Figs. 9 and 10) are complete, and form the main part of the snout. Each maxilla contacts the premaxilla anteriorly, the nasal medially, the lacrimal posteromedially and the jugal posteriorly. The anterior part of the lateral margin is embayed by a recess that is continuous with the similar feature on the premaxilla, forming the posterior part of the notch for the fourth dentary tooth. Posterodorsal to this notch is the more anterior of the two depressions that Young (1982) described on the lateral surface of the maxilla and regarded as a distinguishing feature of P. huiningensis. This depression is shallow, irregular and transversely elongate, extending from just above the alveolar margin to a point on the suture with the nasal. The depression is bounded posteriorly by a low elevation aligned with the large fifth maxillary tooth. The opposite side of the skull is unfortunately damaged in this region, and the snout is distorted by leftward skew. There is at least a slight anterior depression corresponding in position to the one on the right side, but it may not be equally large and well-developed. Even if the anterior depression is a genuine morphological feature rather than an artifact of distortion, it represents little more than an irregularity in the surface of the snout, and its taxonomic value is questionable.

A partial lacrimal is preserved on the right side of the skull. There is no discernable maxillary process either clasped by the lacrimal or projecting between the lacrimal and the nasal. The medial part of the maxilla-lacrimal suture is transverse, but the lateral part of the suture extends posterolaterally and is situated just within a second depression on the maxilla. This posterior depression has the shape of an irregular oval, and is separated from the alveolar margin by a prominent area of bone. The left side of the skull is again damaged in the corresponding region, in that the surface of the maxilla is worn away and the body of the maxilla is interrupted by a slit-like hole that is clearly artifactual. Young (1982) considered the posterior depression to represent a degenerate antorbital fenestra, but this interpretation seems untenable. As with the anterior depression, it is difficult to rule out the possibility that the posterior depression is simply a product of damage and/or deformation, particularly given that the sculpturing inside the depression appears eroded. Furthermore, the region of the crocodylian snout believed to correspond to the formerly open antorbital fenestra is situated mainly on the maxilla, with a small degree of lacrimal participation, but does not extend onto the jugal (Witmer, 1995). However, the right jugal of P. huiningensis does contribute to the posterior depression on the snout. Young compared P. huiningensis with a notosuchian, Uruguaysuchus aznarezi, but the antorbital fenestra in U. aznarezi has no jugal component (Soto, Pol & Perea, 2011). Moreover, members of Crocodylia consistently lack an antorbital fenestra, so the presence of even a degenerate version of this feature in the specimen would be extremely surprising. For these reasons, the posterior depression is clearly not a homologue of the antorbital fenestra. Like the anterior depression, it may represent either an artifact or a genuine topological feature, but seems unpersuasive as a taxonomic character.

The left maxilla bears nine teeth, while the right bears twelve, rather than the eleven reported by Young (1982). Combining information from the two sides of the skull shows that the first two maxillary teeth share the small and slender morphology of the first three premaxillary teeth, while the third is slightly larger. The fourth tooth is even larger than the third, and the fifth and sixth teeth are the largest in the maxilla. The seventh to ninth maxillary teeth are shorter and blunter than the fourth. The tenth to twelfth teeth are slightly larger than the fourth, but smaller than the fifth and sixth.

Lacrimal—Only an anterior remnant of the right lacrimal is preserved (Figs. 9A, 9B, 10A and 10B). A small portion of the left lacrimal may be present as well, but no suture is visible to separate this bone from the maxilla. The right lacrimal contacts the maxilla anteriorly, the nasal medially and the jugal ventrally. The fact that there is a preserved remnant of the lacrimal, but no preserved remnant of the prefrontal, indicates that the former bone extended further anteriorly in the intact snout.

Nasal—Both nasals are preserved (Figs. 9A, 9B, 10A and 10B), but are damaged in the region of the external naris (EN). Each nasal contacts its counterpart medially, the premaxilla anteriorly and the maxilla laterally, and presumably reached the frontal posteriorly. The nasals clearly extended into the EN, forming a long nasal bar, but most of this structure has been obliterated by a transverse gouge across the dorsal surface of the snout. The tips of the two nasal processes remain in place anterior to the gouge, isolated within the narial opening. The nasal-maxilla suture is linear in dorsal view. A very thin, meandering transverse line near the posterior margin of the preserved part of the snout may represent the nasofrontal suture. If this is the case, the contact between the nasal and frontal is broad, and a small remnant of the frontal remains in place.

Jugal—As preserved, the right jugal (Figs. 9A, 9B, 10A and 10B) extends posteriorly somewhat beyond the level of the postorbital bar. A small anterior part of the left jugal may be preserved in the vicinity of the orbit, but this is uncertain. The right jugal contacts the maxilla anteriorly, and the lacrimal dorsally. The base of the slightly inset postorbital bar is not separated by a depression from the lateral margin of the jugal, and bears a small longitudinal crest on its anterior surface.

Lower jaw

Dentary—The left dentary (Figs. 9 and 10) is broken at the level of the tenth maxillary tooth, with only the portion anterior to the break remaining in place. The right dentary is also broken away posteriorly, but remains intact to the level of the anterior part of the external mandibular fenestra (EMF). The dentary contacts the splenial medially and the surangular posterodorsally. The tooth-bearing portion of the dorsal margin of the dentary is similar to its equivalent in Asiatosuchus nanlingensis in undulating gently along its length, whereas the posterior part of the dorsal margin slopes posterodorsally. The ventral margin of the dentary is slightly convex. The dentary symphysis extends posteriorly as far as the level of the fifth dentary tooth. The dentary-surangular suture intersects the anterodorsal margin of the EMF.

The preserved part of the left dentary bears eleven teeth, rather than the seventeen described by Young (1982), while the right bears fourteen teeth. The posteriormost dentary tooth is displaced and rotated, resting in the space between the dentary and maxilla. However, it appears to belong to the dentary because of its lack of the consistently bulbous morphology seen in the posterior maxillary teeth. The anterior two dentary teeth point dorsally, rather than anterodorsally as in some other crocodylians (e.g., Osteolaemus tetraspis (Cope, 1861)). The fourth dentary tooth is the largest, and the third is the second-largest. The eleventh dentary tooth is also enlarged, having twice the diameter of the tenth.

Splenial—The right splenial is preserved, although its surface is mildly damaged (Figs. 9C, 9D, 10C and 10D). The shape of the anterior part of the splenial suggests this bone was likely excluded from the mandibular symphysis, although the anteriormost part of the thin splenial is slightly damaged. The splenial forms the anterior margin of a low, elongate opening in the medial surface of the mandible that may represent the foramen intermandibularis medialis (FIM). The splenial appears to contact anterior processes of the angular both ventral and dorsal to this opening, but the sutural contacts are difficult to follow with certainty, and the angular cannot be traced onto the lateral surface of the mandible. Another putative suture line dorsal to the opening may represent the contact between the splenial and a small exposed strip of the coronoid. There is no evident foramen intermandibularis oralis near the symphysis or on the medial surface of splenial body. A small anteroventral portion of the left spenial is exposed, but the anteriormost part of the bone is missing or damaged.

Surangular—Only the anterior part of the right surangular is preserved, contacting the dentary anteriorly. As in other alligatoroids, the surangular bears a forked process (Figs. 10A and 10B ) whose ventral branch is long, closely approaching the posteriormost preserved dentary teeth. The preserved portion of the surangular represents the tapering ventral process, which extends anteriorly near the dorsal margin of the lateral surface of the mandible. The posterior part of the surangular is damaged, and whether a sulcus is present on the dorsal portion of this bone as in A. nanlingensis cannot be determined.

Angular—Only the anterior part of the right angular is preserved (Figs. 9C, 9D, 10A and 10B), but this bone evidently contacts the dentary anteriorly and the surangular dorsally. The angular comprises the anteroventral part of the EMF, and possibly the ventral part of the FIM.

Phylogenetic Analysis: Methods

To test the proposed status of Eoalligator chunyii as a junior synonym of Asiatosuchus nanlingensis, and to explore the phylogenetic positions of the latter species and Eoallligator huiningensis, a cladistic analysis was conducted using TNT 1.0 (Goloboff, Farris & Nixon, 2003) on a modified version of the matrix of Brochu (2012).

Specifically, five new taxa were added to the matrix: Asiatosuchus nanlingensis and Protoalligator huiningensis were added to explore their phylogenetic positions (Young, 1964; Young, 1982); Krabisuchus siamogallicus and an unnamed alligatorine (the “Maoming specimen”) were added, based on codings from previously published matrices (Martin & Lauprasert, 2010; Skutschas et al., 2014), to evaluate the phylogenetic relationships among Asian alligatoroids; and finally, “Eoalligator chunyii” was added as a separate OTU to test its possible synonymy with A. nanlingensis.

Two new characters were added to the data matrix, as follows. Character 190 pertains to the premaxillary teeth: largest premaxillary tooth is the second (0), the third (1) or the fourth (2). Character 191 (Fig. 11) pertains to the surangular: dorsal surface of the surangular is smooth (0), or bears a large sulcus next to the anterior half of the glenoid fossa (1). The two characters were coded mostly based on published figures (Brochu, 1999; Brochu, 2011; Brochu, 2012), but Asiatosuchus nanlingensis, Alligator sinensis, Eoalligator chunyii, Protoalligator huiningensis, Planocrania datangensis and Planocrania hengdongensis were coded based on direct observation of specimens.

Figure 11 Comparison between the surangular of Alligator sinensis (IVPP 1335) (A) and the surangular of Asiatosuchus nanlingensis (IVPP V 2721.1) (B).

In A. sinensis there is no sulcus on the dorsal surface of the surangular, while in A. nanlingensis a large sulcus is present on the dorsal surface of the surangular. Scale bars = 1 cm.

“E. chunyii”, P. huiningensis and A. nanlingensis were all coded into the matrix separately. Codings for “E. chunyii” were based on the samples of crocodylian material from localities L1 and L3, all of which we consider to represent the same species (see ‘Discussion’). Similarly, codings for A. nanlingensis were based in principle on the crocodylian material from L4, although only the pair of large partial mandibles included in the holotype (IVPP V 2773.1) yielded morphological information that could be coded into the matrix. In total, the modified character matrix consisted of 105 ingroup taxa and 191 morphological characters, about half of which could be coded for both “E. chunyii” and P. huiningensis. Bernissartia fagesii was treated as an outgroup taxon.

A new technology search based on 100 random addition sequence replicates and 1,000 random seeds was implemented using TNT (version 1.1; Goloboff, Farris & Nixon, 2003). The advanced search settings were changed to ensure enough iterations: 100 sectorial search drifting cycles, 100 ratchet iterations, 100 drift cycles and 100 rounds of tree fusion for every replicate. Multistate characters were left unordered. All characters were equally weighted. A standard bootstrap analysis of 1,000 replicates was followed to evaluate the support for each node.

New genus

The electronic version of this article in Portable Document Format (PDF) will represent a published work according to the International Commission on Zoological Nomenclature (ICZN), and hence the new names contained in the electronic version are effectively published under that Code from the electronic edition alone. This published work and the nomenclatural acts it contains have been registered in ZooBank, the online registration system for the ICZN. The ZooBank LSIDs (Life Science Identifiers) can be resolved and the associated information viewed through any standard web browser by appending the LSID to the prefix http://zoobank.org/. The LSID for this publication is: urn:lsid:zoobank.org:pub:2484A449-CB00-4B2B-B6EF-10505CED6AB1. The online version of this work is archived and available from the following digital repositories: PeerJ, PubMed Central and CLOCKSS.

Figure 12 Strict consensus of 193 equally parsimonious trees found in the cladistics analysis (105 ingroup taxa and 191 morphological characters).

Length = 728 steps; Consistency Index (CI) = 0.338; Retention Index (RI) = 0.806. Square brackets mark two nominal species of Eoalligator and Asiatosuchus nanlingensis. Bremer support values and bootstrap values are provided above/below each node show limited support for clades of interests.

Phylogenetic Analysis: Results

The analysis recovered 193 equally parsimonious trees, each 728 steps in length with an ensemble consistency index (CI) of 0.338 and an ensemble retention index (RI) of 0.806. The strict consensus tree is displayed in Fig. 12 with Bremer support indicated for each node. The general topology was consistent with the results of previous studies (Brochu, 1999; Brochu, 2012; Martin & Lauprasert, 2010), but resolution was relatively poor within Crocodyloidea and Alligatoroidea. The bootstrap values suggest very limited number of support for clades of interest, which may result from the incomplete coding of the character matrix.

Within Crocodyloidea, four monophyletic groups were recovered: tomistominae, Osteolaemus, Brochuchus pigotti + Euthecodon arambourgii and Eoalligator chunyii + Asiatosuchus nanlingensis. Within Alligatoroidea, Globidonta emerged as a clade containing several smaller monophyletic groups: Procaimanoidea kayi + Arambourgia gaudryi + Procaimanoidea utahensis; Alligator + Wannaganosuchus brachymanus; Paleosuchus; Melanosuchus + Caiman latirostris + Caiman lutescens; and Caiman crocodilus + Caiman yacare.

The two putative species of Eoalligator did not form a monophyletic group: Eoalligator chunyii was recovered within Crocodylidae as the sister-taxon of Asiatosuchus nanlingensis, a placement consistent with the absence in specimens assigned to E. chunyii of one important derived feature normally seen in alligatoroids (character 70: medially positioned foramen aerum). On the other hand, Protoalligator huiningensis was placed in a polytomous Globidonta, suggesting that this species represents an alligatoroid of uncertain affinities. To move P. huiningensis outside Alligatoroidea required four additional steps, suggesting this species is indeed an alligatoroid as indicated by the long surangular process. Recovering a monophyletic Eoalligator required an additional three steps, and recovering a monophyletic Eoalligator within Alligatorinae as suggested by Young (1964) and Young (1982) required an additional eleven steps. Although A. nanlingensis and “E. chunyii” were posited as a monophyletic group at the base of Crocodylidae in the cladogram, only one additional step was needed to move this small clade to the base of Crocodyloidea. On the other hand, to move the A. nanlingensis + “E. chunyii” clade into Alligatoroidea required five steps, and to move it to the base of Brevirostra required four steps. Support for the sister-group relationship between A. nanlingensis and “E. chunyii” was relatively weak, with only 1 additional step being required to separate these putative species, but the non-monophyletic constraint caused Crocodylia to collapse. Importantly, the results of the analysis are consistent with the interpretation that E. chunyii is a synonym of A. nanlingensis. The results also provide substantial evidence that A. nanlingensis belongs to Crocodyloidea, if not Crocodylidae, but the exact position of this species within Crocodyloidea may change when more intact specimens are found.

Asiatosuchus was not obtained as a monophyletic group. Although all three Asiatosuchus species fell within Crocodyloidea, the Mongolian species A. grangeri (the type species of Asiatosuchus) and the European species A. germanicus were positioned well basal to A. nanlingensis and were not recovered as sister-taxa. The lack of a monophyletic Asiatosuchus in our phylogenetic results, combined with the fact that Young’s (1964) original justification for assigning A. nanlingensis to Asiatosuchus relied mainly on vague similarities to A. grangeri, raises the strong possibility that A. nanlingensis will ultimately need to be moved to a new genus. However, this taxonomic step will require detailed comparisons between A. nanlingensis and A. grangeri, ideally in the context of a comprehensive review of Asiatosuchus, and therefore lies beyond the scope of this paper. Accordingly, we take a conservative approach and continue to refer to A. nanlingensis under the genus name Asiatosuchus.

Discussion

Asiatosuchus nanlingensis and Eoalligator chunyii were named in the same paper (Young, 1964) based on material discovered at localities within the Nanxiong Basin that we designate L1–L6, and one locality (L3) even yielded putative specimens of both taxa. Despite a distinct size difference between specimens assigned to A. nanlingensis and those assigned to E. chunyii, mandibles of the two putative taxa share four significant morphological characters: (1) dorsally positioned sulcus within the surangular, which is unique to these two nominal species; (2) surangular-articular suture that extends anteroposteriorly within the lateral hemi-fossa of the glenoid (shared by various crocodyloids); (3) posterodorsally pointing retroarticular process; and (4) gently curved mandible. Further anatomical comparisons are limited by the fact that the holotype of A. nanlingensis (IVPP V2773) consists only of a pair of partial mandibles, combined with a few postcranial fragments. This specimen represents the entire sample of crocodylian material from the site L4. Based on their shared provenance with the diagnostic mandibles, the postcranial fragments probably do belong to A. nanlingensis, even though two vertebrae are too small to belong to the same individual as the rest of the holotype. However, the identification of the postcranial specimens originally assigned to A. nanlingensis is something of a moot point, considering that they provide little taxonomically relevant information in any case.

Only three of the other localities (L1, L3 and L5) yielded significant samples of crocodylian material, although L2 and L6 each produced an isolated, taxonomically indeterminate partial crocodylian mandible. L1 produced a large collection of crocodylian bones which, combined with a few non-crocodylian fragments, were designated by Young (1964) as the holotype of Eoalligator chunyii(IVPP V2716). This material includes three partial mandibles, one of which (part of IVPP V2716-3) is a posterior fragment that displays the distinctive surangular sulcus of A. nanlingensis and also resembles the holotype of this species in the morphology of the retroarticular process and surangular-articular suture. No important anatomical differences, other than size, exist between IVPP V2716-3 and the holotype of A. nanlingensis. The other bones and bone fragments that make up IVPP V2716 are less directly comparable to the holotype of A. nanlingensis, even the other mandibular fragments being anterior ones that do not preserve the key diagnostic features of the articular region. However, there are no anatomical discrepancies in the IVPP V2716 material that would suggest the presence of more than one crocodylian taxon at L1. The two posterior skull roof fragments from this site (IVPP V2716-1.1 and V2716-13) both display an unusual longitudinal sulcus between the two supratemporal fenestrae, suggesting that at least these two partial skulls are conspecific. Given the high level of morphological consistency in the L1 sample, the diagnostic similarities and lack of diagnostic differences between IVPP V 2716-3 and the holotype of A. nanlingensis, and the stratigraphic and geographic proximity of L1–L4, we refer all of the crocodylian material from L1 (i.e., the crocodylian component of IVPP V 2716) to A. nanlingensis, the senior synonym of E. chunyii based on page priority (Young, 1964).

The sample of crocodylian material from L3 includes three mandibular fragments, one of which (IVPP V 2721.1) is from the posterior end of the lower jaw and preserves diagnostic features of A. nanlingensis. The other mandibular fragments (IVPP V 2721.5 and IVPP V 2721a) are from the anterior part of the lower jaw. IVPP V 2721.5 is poorly preserved, but is morphologically compatible with the holotype of A. nanlingensis and similar to anterior lower jaw fragments from L1. Although very small and lacking alveoli, IVPP V2721a shows lateral sculpturing similar to that seen in the holotype of A. nanlingensis. We refer IVPP V 2721.5 and V 2721a to A. nanlingensis based on these comparisons, and based on their association with IVPP V 2721.1. A caudal vertebra from the same locality (IVPP V 2721.4) may belong to A. nanlingensis as well, on the basis of provenance, but this identification is tentative. The fragment IVPP V 2721.1 is particularly significant, because it represents the only Cretaceous specimen that shows the diagnostic posterior mandibular features of A. nanlingensis and therefore constitutes the key evidence that A. nanlingensis survived the end-Cretaceous extinction.

The crocodylian collection from L5 is the only one considered in this paper that preserves clear evidence for the presence of multiple taxa. This sample contains a partial dentary (IVPP V 2772.1) that resembles specimens assigned to A. nanlingensis, including the partial dentaries of the holotype. However, the sample also includes a longirostral mandibular symphysis (IVPP V 2772.2) that clearly belongs to a different species, in addition to a number of isolated teeth with serrated carinae (IVPP V 2772.3) that probably belong to a third species. Other bones from this site are individually undiagnostic, and most could potentially belong to any of the three taxa. The partial dentary IVPP V 2772.1 probably represents A. nanlingensis, but other specimens from L5 cannot be even tentatively referred to this species.

The cladistic analysis carried out for this study recovered “E. chunyii” and A. nanlingensis as sister taxa within Crocodylidae (Fig. 12). Their sister-group relationship was supported by two of the morphological characters listed above: a sulcus within the surangular (character 73), and an anteroposteriorly oriented surangular-articular suture (character 74). The posterodorsally directed retroarticular process is shared with most crocodylians, and the gently curved mandible is shared with many members of Brevirostres. Furthermore, there are no clear morphological differences between A. nanlingensis and “E. chunyii”. In the light of the fact that the two species are essentially indistinguishable in distribution and morphology, the results of the cladistic analysis corroborate the status of E. chunyii as a junior synonym of A. nanlingensis. Because “E. chunyii” is the type species of Eoalligator, the new genus name Protoalligator is required to accommodate “E.” huiningensis as Protoalligator huiningensis.

Given that P. huiningensis lies well outside Alligator, the apparent presence of a complete nasal bar that bisects the external naris (EN) is intriguing. To move P. huiningensis within Alligator, maintaining the nasal bar as a synapomorphy exclusive to this genus among alligatoroids, requires only three additional steps. However, placing P. huiningensis within Alligator would be inconsistent with the presence of certain plesiomorphic characters in this species (lacrimal extends further anteriorly than prefrontal, notch for fourth dentary tooth present between premaxilla and maxilla). Additional fossils that provide morphological information regarding the posterior part of the skull, and regarding taxonomically important postcranial bones such as the scapula and coracoid (Brochu, 1995) have the potential to further improve our knowledge of interrelationships within Alligatoroidea.

Based on the taxonomic revision presented in this paper, four alligatoroid species are currently known in the Chinese fossil record: Protoalligator huiningensis from the Paleocene of Anhui; Alligator luicus from the Miocene of Shandong; Alligator sinensis, which may date back to the upper Pleistocene (Shan, Cheng & Wu, 2013); and the Maoming specimen from the Eocene of Guangdong (Skutschas et al., 2014). Brief comparisons among these taxa reveal some interesting patterns of character distribution. In P. huiningensis, A. luicus, and the Maoming specimen, the lacrimal extends further anteriorly than the prefrontal, a plesiomorphic feature. By contrast, most non-caimanine alligatorines share the derived condition, in which the prefrontal extends further anteriorly. This is the case in Allognathosuchus, Wannaganosuchus brachymanus, Procaimanoidea, Arambourgia gaudryi, and most species of Alligator.

An additional plesiomorphy seen in P. huiningensis and the Maoming specimen is the absence of a process of the maxilla either clasped by the lacrimal or intruding between the lacrimal and the prefrontal. In A. luicus, a small flange of the maxilla is present between the lacrimal and the prefrontal. In A. sinensis the maxillary process is clasped by the lacrimal, a derived feature also seen in A. mississippiensis. Globidontans also have a maxillary process, but the position of this feature varies within the group. The maxillary process is positioned between the lacrimal and the prefrontal in a few stem-alligatoroids, including Stangerochampsa mccabei, Albertochampsa langstoni, and Brachychampsa. In more derived alligatoroids, the maxillary process is clasped by the lacrimal. The distribution of this character state suggests that the maxillary intrusion may have first appeared as a process situated between the lacrimal and prefrontal, and subsequently shifted in more derived forms to lie fully within the lacrimal. However, this scenario cannot be confirmed at present, because of the poorly resolved phylogeny.

The absence of a complete nasal bar is a plesiomorphic character shared by nearly all alligatoroids other than Alligator. The Maoming specimen seems to retain this plesiomorphy, although preservation is poor near the EN (Skutschas et al., 2014). In A. luicus and A. sinensis, the external naris (EN) is bisected by a complete nasal bar formed by posterior processes of the premaxillae and anterior processes of the nasals, as in other species of Alligator. In P. huiningensis the premaxilla clearly bears a posterior process, but part of this structure is broken away. As a result, it is impossible to determine the original length of the posterior process, or whether it contacted the nasal process to form a complete nasal bar. If a nasal bar was indeed present, P. huiningensis would presumably have acquired this highly derived feature independently from Alligator. Even if the nasal bar was incomplete, however, the posterior process of the premaxilla would still represent a highly unusual feature among Paleocene alligatoroids, and a case of convergence between P. huiningensis and Alligator.

Unfortunately, the results of the cladistic analysis are rather poorly resolved. The node corresponding to Crocodyloidea (Fig. 12) is a polytomy, so this cladogram does not provide a clear framework for investigating crocodyloid evolution and paleobiogeography. Nevertheless, the Cretaceous—Paleocene age of A. nanlingensis has interesting potential implications. Prodiplocynodon langi from the Cretaceous of North America (Mook, 1941) is the oldest known crocodyloid, but is not much older than A. nanlingensis. If A. nanlingensis is indeed a crocodylid as weakly posited by our phylogenetic analysis, rather than merely a basal crocodyloid, the short temporal gap between P. langi and A. nanlingensis implies a rapid series of divergences within Crocodyloidea. However, more time will be available for these divergences if older crocodyloids that are more basal than P. langi are discovered in the future. If, on the other hand, A. nanlingensis is a basal crocodyloid, its provenance suggests Asia was the setting for some portion of early crocodyloid evolution.

The phylogeny (Fig. 12) also indicates that there must have been at least two historical dispersals of alligatoroids into China, the first involving ancestors of P. huiningensis and the second involving ancestors of A. sinensis. The first dispersal may also explain the presence of the Maoming specimen in the Eocene of Guangdong, and of Krabisuchus in the Eocene of Thailand. The existence of the Chinese Miocene taxon Alligator luicus may imply a third dispersal, depending on the phylogenetic relationships between this species and other Asian alligatoroids.

P. huiningensis is known only from the middle Paleocene Wanghudun Formation of Anhui (Young, 1982). The only known fossil specimen that can be securely attributed to A. sinensis is probably Late Pleistocene in age (Shan, Cheng & Wu, 2013), although there is also a poorly dated possible record from the Neogene of Thailand (Claude et al., 2011) and a fragmentary, taxonomically questionable one from the Pliocene of Japan (Iijima, Takahashi & Kobayashi, 2016). Accordingly, the first dispersal must have occurred no later than the middle Paleocene, whereas the second must have occurred no later than the Pleistocene. One possible route by which the ancestors of P. huiningensis could have migrated to China led west from North America, via the Bering Land Bridge. This land bridge facilitated many terrestrial faunal exchanges between Asia and North America during the late Paleocene and early Eocene Beard, 2002; Smith, Van Itterbeeck & Missiaen, 2004; Sole & Smith, 2013, a time interval characterized by the globally elevated temperatures of the Paleocene–Eocene Thermal Maximum (PETM) event. P. huiningensis is from the Wanghudun Formation, which appears to correlate with the upper part of the Shanghu Formation and the lower part of the Nongshan Formation in the Nanxiong Basin (Missiaen, 2011). The Shanghu Formation has produced mammalian species belonging to various groups, such as mesonychids and carnivorans, which are shared with North and South America (McKenrta & Bell, 1997). However, these taxa are insufficiently well known to provide a strong basis for biogeographic interpretations (Missiaen, 2011). The Nongshan Formation contains mammals such as arctostylopids and Ernanodon, but the presence of these taxa is no longer considered to represent evidence for early Paleocene migrations between Asia and the Americas (Missiaen et al., 2006; Rose, 2006). As a result, the mammalian species currently known from the early Paleocene of China are now considered to belong to endemic groups (Missiaen, 2011; Ting et al., 2011). However, a recent report on a spadefoot toad from the upper Paleocene of Mongolia suggests that members of this clade migrated from North America to Asia at some point prior to the larger-scale mammalian faunal exchange of Paleocene–Eocene boundary times (Chen et al., 2016). Given that P. huiningensis is middle Paleocene in age, its ancestors must have come to Asia no later than the early middle Paleocene. This datum also suggests a migration from North America to Asia predating the mammalian exchange, but whether P. huiningensis and the Mongolian spadefoot toad dispersed in the same event remains unknown. Similarly, the ziphodont planocraniid crocodylians of the Paleogene presumably originated in Laurasia, being known from North America and Eurasia (Brochu, 2013). They must have dispersed at least once between these continents, but whether this dispersal coincided with that of P. huiningensis or its ancestors is unclear. The driving factors behind the dispersal(s) of all these non-mammalian taxa are equally uncertain.

Given that at least some basal alligatoroids presumably shared the lack of salt tolerance seen in extant taxa, Paleocene alligatoroid dispersals may have been restricted to land corridors. The ancestors of P. huiningensis might have reached Asia by dispersing eastward from North America through Europe, rather than westward across the Bering Land Bridge. However, the mammalian fossil record suggests that the direct terrestrial faunal exchanges between Asia and Europe that have so far been documented did not begin until around the Paleocene–Eocene boundary (Godinot & Lapparent de Broin, 2003; Sole & Smith, 2013). Nevertheless, the presence of the Asiatosuchus-like taxon Diplocynodon in Europe indicates that this pathway may have been viable for crocodylians, despite being considerably longer than the alternative westward route (Martin et al., 2014). Resolving the taxonomic problems within Asiatosuchus could be helpful in clarifying whether crocodylian faunal exchanges took place between Europe and Asia before the PETM. The Turgai Strait would have posed an obstacle to dispersal via this pathway for at least part of the Paleocene. On the other hand, sea level regressions of the Tethys Sea may have allowed terrestrial vertebrates to migrate during certain time intervals within Paleocene, so that the Turgai Strait may not have represented an absolute barrier to movement between Asia and Europe. As a result, both dispersal routes remain plausible given current knowledge of the phylogenetic relationships and geographic distribution of Paleogene crocodylians. Regardless of the route taken, the dispersal of the ancestors of P. huiningensis represents an important biogeographic event predating the Eocene dispersal inferred previously (Martin et al., 2014).

Conclusions

(1) “Eoalligator chunyii” is a junior synonym of Asiatosuchus nanlingensis.

(2) Given that the cost to move Asiatosuchus nanlingensis outside Crocodyloidea in our phylogenetic analysis was relatively high (four steps), A. nanlingensis is most likely to be a crocodyloid if not a crocodylid.

(3) Given that the cost to move Protoalligator huiningensis outside Alligatoroidea in our phylogenetic analysis was relatively high (four steps), P. huiningensis is most likely to be an alligatoroid. However, this species both retains several primitive alligatoroid characters and probably exhibits an advanced character otherwise limited to Alligator among Alligatoroidea (a complete nasal bar), and its affinities within Alligatoroidea are uncertain.

(4) The ancestors of Protoalligator huiningensis must have migrated to Asia by the early middle Paleocene. This dispersal probably took place via the Bering Land Bridge, although dispersal to Asia through Europe may represent a plausible alternative.

Supplemental Information

Supplemental Information 1 Character coding of new taxa including: “Eoalligator chunyii”, Protoalligator huiningensis, Asiatosuchus nanlingensis, Krabisuchus siamogallicus and Maoming Specimen; character coding of modified character (74) and new character (190).

Click here for additional data file.

We are grateful to Fu Hua-lin for carrying out further preparation on some historic specimens, Li Lu and Tong Hai-yan for identifying turtle remains mixed in with the crocodylian material examined in this paper, Tom Stidham for advice on Paleogene faunal dispersals, and Jess Miller-Camp for providing literature. We also thank Jess Miller-Camp, Francisco Ortega and Jeremy Martin for thoughtful reviews that substantially improved the paper.

Institutional abbreviations

IVPP Institute of Vertebrate Paleontology and Paleoanthropology, Chinese Academy of Sciences, Beijing, China.

Additional Information and Declarations

Competing Interests

Author Contributions

Data Availability

New Species Registration

The authors declare there are no competing interests.

Yan-yin Wang conceived and designed the experiments, performed the experiments, analyzed the data, contributed reagents/materials/analysis tools, wrote the paper, prepared figures and/or tables.

Corwin Sullivan conceived and designed the experiments, performed the experiments, analyzed the data, contributed reagents/materials/analysis tools, wrote the paper, reviewed drafts of the paper, provide valid suggestions in wording and organization of the manuscript.

Jun Liu conceived and designed the experiments, performed the experiments, analyzed the data, contributed reagents/materials/analysis tools, wrote the paper, reviewed drafts of the paper, provide valid suggestions in wording and help in processing the figures.

The following information was supplied regarding data availability:

The raw data has been supplied as a Supplemental Dataset.

The following information was supplied regarding the registration of a newly described species:

Publication LSID: urn:lsid:zoobank.org:pub:2484A449-CB00-4B2B-B6EF-10505CED6AB1

Genus LSID: urn:lsid:zoobank.org:act:64EDF42C-5096-489C-BF93-F4462C886D67

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
