# Peer review of "Taxonomic revision of Eoalligator (Crocodylia, Brevirostres) and the paleogeographic origins of the Chinese alligatoroids"

_PeerJ, doi:10.7717/peerj.2356_

## Round 0.1 · original submission · Major Revisions

Dear authors,

As the two reviewers have not come to a common decision, I have decided upon 'major revisions'. That being said, please pay close attention to both reviewers comments.

·

Basic reporting

Lines 60–62: There are also alligatoroids from Thailand (Martin and Lauprasert, 2010; Claude et al. 2011), so it might be better to push the importance of Chinese alligatoroids by saying that, given the multiple recent discoveries in East and Southeast Asia, we are only beginning to sample Asian alligatoroid diversity.

Lines 66–67 Brochu (1999) actually says that *none* of the living alligatorids have salt glands. I think there may have been one study that claims to have found them in A. sinensis, but they were rudimentary and non-functional. There’s a good review of osmoregulation in crocodylians in Grigg and Kirshner (2015). It’s more complicated than just lingual glands. There are also glands in the cloaca that reabsorb water from urine, crocodile kidneys handle salt a bit better, they instinctively don't drink salt water while alligators have to learn not to, and crocodile skin is more resistant to osmosis.

Line 186–187: There's no precedent for the recess doing otherwise than accommodating the fourth dentary tooth, so it's safe to say it would have done so rather than just be likely to.

Lines 202–204: The lack of a maxillary process is plesiomorphic, so it might be better to say what groups do have it (derived gavialoids, New World Crocodylus and C. niloticus, a couple autapomorphic crocodyloids, most tomistomines, most alligatoroids).

Lines 203–204: Paleosuchus is a caimanine.

Lines 217–219: Same here. The groups with the derived condition of a shorter lacrimal are: nettosuchids and most alligatorines.

Lines 280–281: Same here. The groups with the derived condition of a more medio-dorsally placed spine are: a couple autapomorphic crocodyloids, some tomistomines, and alligatorines.

Lines 292–293: This character is also present in Deinosuchus, Diplocynodon, and Brachychampsa (i.e., it's synapomorphic for the node sister to Brachychampsa).

Line 342: I'm not familiar with many species outside Crocodylia, but this feature is true for all the non-eusuchian neosuchians I've seen. Probably best to just remove the phrase "as in extant crocodylians".

Lines 386–387: The two species don't even come close to co-occurring, so this statement seems unnecessary.

Lines 438–439: Given that this is a unique feature, and perhaps one of the features you use to justify synonymization, it should be emphasized in a figure. Outlining it with a dotted line similar to the solid lines you've highlighted sutures with would be sufficient.

Lines 441–442: This feature is in all crocodyloids but not in Alligator or Gavialis.

Lines 569–576: Have you considered that the serrated teeth may be specimens of Planocrania? There are two species of this ziphodont eusuchian, both from the Paleocene of China (hengdongensis [Li, 1984] and datengensis [Li, 1976]).

Lines 672–673, Figure 7: If you are unsure about these sutures, they should be depicted as dashed lines rather than solid (as you've done for other sutures where surficial bone has been removed). I especially encourage this as that would be an unusual shape for a crocodylian nasofrontal suture.

Lines 719–720: Since there were only two changes, it's probably worth actually stating what they are here (and figuring them) rather than forcing the reader to switch over to the supplemental materials to find them.

Line 763–764: Since the genus was not recovered as monophyletic, it's worth stating which is the type species.

Line 796: Better to say "plesiomorphic" than "primitive". "Primitive" has some unfortunate connotations in English. It encourages the inaccurate ladder-like line of thought that evolution moves towards "inherently better" rather than simply "better for the current state of things". It's especially problematic in that it reinforces the false idea of "living fossils" being a real phenomenon. But that's a whole other rant I won't bother you with here.

Lines 812–837: Compare with the above comments regarding character state distributions to ensure internal consistency.

Line 824: Replace "alligatoroids that are more advanced than the diplocynodontine" with "globidontans".

Line 884: You have an incomplete sentence here.

Line 906: You haven't discussed the physiological restrictions to saltwater dispersal in alligatorines beyond mentioning the existence (or lack thereof) of salt glands. You should include a small discussion of it in the introduction or discussion before bringing it up in the conclusions.

Figure 9: A couple typos; should be Maoming *specimen* and Alligator *mefferdi*.

References cited above:
Claude, J., W. Naksri, N. Boonchai, E. Buffetaut, J. Duangkrayom, C. Laojumpon, P. Jintasakul, K. Lauprasert, J. Martin, V. Suteethorn, and H. Tong. 2011. Neogene reptiles of northeastern Thailand and their paleogeographical significance. Annales de Paléontologie 97:113–131.
Grigg, G. and D. Kirshner. 2015. Biology and Evolution of Crocodylians. CSIRO Publishing, Clayton South VIC, Australia.
Li, J.L., 1976, Fossils of Sebecosuchia discovered from Nanxiong, Guangdong. Vertebrata PalAsiatica 14(3):169-173
Li, J.L., 1984, A new species of Planocrania from Hengdong, Hunan, Vertebrata PalAsiatica 22(2):123-133.
Martin, J. E. and K. Lauprasert. 2010. A new primitive alligatorine from the Eocene of Thailand: relevance of Asiatic members to the radiation of the group. Zoological Journal of the Linnaean Society 158:608–628.

Experimental design

Figure 9: I strongly discourage the use of a majority-rule consensus in parsimony analyses. They can be very tempting because of their perfect resolution, but that resolution can be a result of methodological bias and doesn't necessarily reflect the best current guess of evolutionary patterns, much less the true topology. (Sumrall et al. 2001). Best to be conservative about that. The strict consensus should be presented for all parsimony-based phylogenetic analyses. If you want to show a second consensus tree that's more resolved, you can use a semi-strict (sometimes called Adam's) consensus, but note that the purpose of this type of consensus is to isolate labile taxa responsible for resolution loss and it should not be treated as a phylogenetic hypothesis in and of itself.

References cited above:
Sumrall, C.D., Brochu, C.A., Merck, J.W., 2001, Global lability, regional resolution, and majority-rule consensus bias, Paleobiology vol. 27 iss. 2, p. 254–261.

Validity of the findings

Line 187: What makes it partly concealed instead of visible, as with the notch of extant crocodylines?

Line 593: I'm not convinced that the nasal bar is complete. It looks to me like it only extends a little over halfway into the naris. Even if there are small premaxillary processes pointing backwards (and there seems to only be the barest hint of one), consider that the space between them could have been spanned by cartilage which never ossified. It might be worth looking for soft tissue descriptions of modern crocodylines to see if their nares are bisected by cartilage. I would imagine that would be the case.

Lines 698–699: Not necessarily. Take a look at the anterior splenial borders of crocodyloid splenials. Many of them have that sort of shape, and it's that way in modern specimens which are definitely whole.

Lines 729–730: Given the information presented, I can't tell that Gavialoidea and Crocodyloidea have introduced any "noise". Perhaps this is more apparent when the strict consenses of both analyses are compared rather than the majority-rule of the second?

Line 743: Globidonta is sister to Diplocynodontinae, rather than including it, so it's returned as monophyletic here. Did you mean Alligatoridae? Keep in mind that, with polytomies, you aren't *not* recovering clades as monophyletic. Monophly is still a possibility if at least one MPT recovers it as such.

Line 750: I think you meant Globidonta rather than Alligatorinae, if the topology of the majority rules consensus is comparable with that of the strict consistent.

Line 791: Continuing to use a genus name when the type species has been made invalid requires a case ruling by the ICZN; it can't be reassigned by individual authors. An example of a case where continued use of a genus name with a new type species was upheld is here: http://iczn.org/content/case-3515-rhynchotherium-falconer-1868-mammalia-proboscidea-proposed-conservation-usage-desi

Line 852: Nothing in your results or prior discussion suggests that A. nanlingensis could be a basal tomistomine.

Line 860: Be careful about getting too specific with ancestral areas. Keep in mind that the first place we observe a taxon is not necessarily the place it evolved. This is especially relevant for Paleogene taxa in Europe given that relatively few fossiliferous strata from this age are preserved aside from geographically-limited lagerstätten, and their distribution is absolutely not random thanks to the glaciers scraping everything in higher latitudes away. That last bit is especially relevant for alligatoroids, given that they likely arrived in Europe through high-latitude terrestrial corridors.

Figure 7: I disagree with the placement of the premaxilla-maxilla border in C. The line interpreted as a suture in the section where the surficial bone has been scraped away looks like a thin crack to me, unlike the wider, deeper sutures visible around the nasals. It's more in keeping with other crocodylines for the suture to be in the notch that’s just above the 4th dentary tooth (as is more visible in D).

Additional comments

Line 921: My last name's Miller-Camp and I go by Jess.

·

Basic reporting

The manuscript refers to an interesting collection of crocodiles from the Chinese Upper Cretaceous and the Paleogene and gets clarify aspects of some taxa as Eoalligator chunyii, Eoalligator huiningensis y Asiatosuchus nanlingensis. I think that document and analyze this particularly bad known material is interesting. However, I have some problems with the development of the manuscript that perhaps the authors can specify whit more detail.

Experimental design

no comments

Validity of the findings

On Asiatosuchus nanlingensis (plus Eoalligator chunyii)
It seems clear that some of the material originally assigned to Eolligator chunyii has shared characters with Asiatosuchus nanlingensis.
In the proposal, it appears that the most informative elements of the holotype of Asiatosuchus nanlingensis (+ Eoalligator chunyii) are a pair of very large incomplete mandibles (IVPP V 2773.1). However, I am not convinced that all remains assigned to the holotype of Eoalligator chunyii can be considered as belonging to the same species. If all remains don’t belong to a single species, all discussion must be based just on the mandibular characters. In this sense, the association with some ziphodont teeth seem also indicate the need of comparing some elements with those of ziphodont crocodiles as Planocrania.
Finally (as shown clearly the cladogram), there is no relationship between Asiatosuchus grangeri and Asiatosuchus nanlingensis. If this relationship does not exist, I do not understand why it keeps both in the same genus.

On Eoalligator huiningensis
I think there is a major problem with the diagnosis: No autapomorphies, the character combination is very sparse, and the proposed characters are highly incongruent. Under these conditions I do not think they can defend the validity of the especies.
On the other hand, I'm not absolutely sure, but, as the type species of Eoalligator, E. chunyii, is now invalid, if diagnosis were possible, I think that it would be necesary designate a new genus for "E. huiningensis".

On cladistic analyses
Results of both cladistic are rather poorly resolved. That is normal, since the information available is scarce. The two analyses may have an interest to approximate the classification of the two taxa, but I think not serve to base evolutionary and paleobiogeographic proposals. In the latter case, the conclusions based on the results of the phylogeny are highly speculative

On conclusions
About conclusion 1, it seems to be clear that there is a clear relationship between the mandible of Asiatosuchus nanlingensis and some material referred to Eoalligator chunyii. About conclusions 2, 3 and 4., I think the low resolution of the results makes the support of the conclusions is too weak

---

## Round 0.2 · Minor Revisions

Dear authors,

I have made the decision of 'minor revisions' based on the reviewer comments and recommendations. However, the re-analysis mentioned by reviewer one will need to be made.

Please note that reviewer two from the previous round could not review the revised submission. Thus a new reviewer (#3) was asked to review this round.

I look forward to receiving your revised manuscript.

·

Basic reporting

No comments.

Experimental design

Surangular sulcus: I didn’t catch this in my previous review, but the apomorphic sulcus you see on the surangular of Asiatosuchus and “E.” chunyii is not homologous with the sulcus in the character to which you’ve added it as a new state. Brochu’s sulcus is a valley that runs along the lingual surface, with the suture in the nadir. It’s present in most tomistomines and gavialoids. Otherwise, the surangular sits flush against the articular.
Which means the analysis will need to be rerun with the presence/absence of a sulcus on the dorsal surface of the surangular as a completely new character. I don’t imagine this would change the topology of their phylogenetic hypothesis much, if at all, but the character isn’t publishable as is.
This statement is repeated in the attached file with pictures of each state.

Validity of the findings

No comments.

Additional comments

The authors have addressed my previous concerns. I don't foresee making many edits or comments on an additional review, but have selected "major revision" given the need to rerun the analysis.

·

Basic reporting

please see below

Experimental design

please see below

Validity of the findings

please see below

Additional comments

This is an interesting and well-conducted study that clarifies the taxonomic content of several taxa originally described from the Paleocene of China by Young. I am only entering the review process, but judging from the author’s responses and changes applied to this manuscript, I think the authors have appropriately revised their work. This is a signed review and if needed, the authors can contact me (jeremy.martin@ens-lyon.fr). I have some minor comments (structure, references, small typos) and if those could be addressed, I would recommend this manuscript for publication. In any case, this taxonomic and descriptive update will be a useful reference for future studies on the origin and evolution of crocodyloids and alligatoroids.

Most of my comments appear on a .doc version of the ms using track changes and I invite the authors to carefully go through them.

If I understand well the description, the authors seem to confuse the posttemporal canal (which opens on the occipital margin of the skull, just below the skull roof) and the orbitotemporal foramen (which opens on the posterior wall of the supratemporal fenestra. This should be double-checked throughout the text and in the datamatrix.

The sulcus on the surangular seems to be a useful character to associate A. nanlingensis with specimens previously referred to E. chunyii. However, this character cannot be checked in Protoalligator and it should be underlined. Moreover, a comparable sulcus or fossa is also known in Diplocynodon (e.g. Fig. 8A in Martin et al. 2014, ZJLS). How is this affecting, if it does, the proposed phylogenetic framework?

Perhaps the comparison of A. nanlingensis would benefit from previous descriptions of Asiatosuchus-like taxa from Europe, notably on the morphology of the skull table; presence/absence of surangular sulcus, other features (e.g. Berg, 1966; Delfino and Smith, 2009 ZJLS).

Toward the end, issues on biogeography are discussed. The authors may refine some parts of this discussion. Here, I recommend referring to Martin and Lauprasert (2010, ZJLS) and Martin et al. (2014, ZJLS) where biogeographic hypotheses are discussed at length and where other relevant references could also be consulted and cited if judged appropriate.

Figures

The figures are of high quality and the plates are conveniently arranged providing a useful match with the description. In some cases, line drawings are directly marked on the images, leaving little possibility for the reader’s own observations. I let the authors or the editor decide whether a separate interpretative drawing should be provided or not in such cases (maybe in the Protoalligator plate?).

Figure 1 would benefit from an inset map of China or lat.-long. coordinates.

Fig. 2D should be rotated 90° anti-clockwise. The median suture of the palatine should be drawn in Fig. 2I.

Figure 6 is a bit dark in some cases. The inset of image B (showing the surangular sulcus) is difficult to compare with Fig. 4M. This is an important character and I recommend improving the contrast on Fig. 6B.

Close ups of the dentition should be provided if possible for both taxa.

I have noticed the presence of unwanted pixels surrounding some images in the figure plates. Some figures need a final cleaning before they can be ready for publication (e.g. left of A and E and above P in Fig. 2; bottom of E in Fig. 3, etc…).

Jeremy Martin

---

## Round 0.3 · Minor Revisions

Dear authors,

I have given the decision of 'minor revision' for your manuscript. This is because, even though the reviewer 'accepted' the revision, they found some minor issues which should be rectified prior to final acceptance.
I do not foresee any difficulty in you making these final changes.

I have some additional comments that the authors should address prior to resubmission (in addition to those made by the reviewers):

1. Authority and date should be provided for each species-level taxon at first mention. Please ensure that the nominal authority is also included in the reference list.
2. Please replace 'trees' with 'cladograms' or 'topology' where appropriate (i.e. resultant cladograms and consensus topology).
3. Note that the CI is the ensemble consistency index, and the RI is the ensemble retention index. Please update these in the manuscript.

·

Basic reporting

Just noticed that the title has Eoalligator placed in Alligatorinae, though its placement within Globidonta is unresolved in the strict consensus and the text references an additional 11 steps needed to place it inside Alligatorinae. Perhaps the title should say Alligatoroidea instead?

On a related note, in the systematic paleontology section, more nested clades are usually given than only one above genus-level. Crocodylia (Owen, 1842) and Globidonta (Brochu, 1999) could be added.

Use of hyphen instead of en dash: There are still cases of the hyphen needing to be replaced with an en dash where the intended meaning is “though”. Incorrect use noted in references to multiple related localities, references to multiple parts of a figure, and in page numbers in reference list.

Two lines above “phylogenetic analysis:methods”: Typo: anter_i_oventral

First paragraph in “phylogenetic analyses: results”: Change “The boostrap values suggest a very limited number of support….” To “The boo_t_strap values suggest very limited support…”

Specimen numbers: Captions of figures 9 and 10 and Table 1 have no space in the specimen number between “V” and the numerals. Other figures and mentions in-text have a space. Which is correct?

Various figure captions: Use or lack thereof of spaces around the equals signs is not consistent. You mostly have it with only a space after, but there are two instances of a space only before (in Figure 12 caption) and another of a space to either side (Figure 2).

Table 1 caption: Should read “…with detail_ed_ information…”

Experimental design

No comments.

Validity of the findings

No comments.

Additional comments

No substantive changes needed.

---

## Round 0.4 · accepted · Accept

Dear authors,

Many thanks for your revised manuscript. After reading it, I have accepted it for publication in PeerJ.

Once again, thank you for submitting your manuscript to PeerJ and I hope you will use us again as your publication venue.

If we need to clarify any details required to move the manuscript forward, then our production staff will get in touch with you. Otherwise, a proof will be forthcoming shortly for your review.

Congratulations and thank you for your submission.